# AIR QUALITY PREDICTION WITH PHYSICS-GUIDED DUAL NEURAL ODES IN OPEN SYSTEMS

**Jindong Tian[1], Yuxuan Liang[3], Ronghui Xu[1], Peng Chen[1], Chenjuan Guo[1],**
**Aoying Zhou[1], Lujia Pan[2], Zhongwen Rao[2], Bin Yang[1]***
[1]East China Normal University, [2] Huawei Noah's Ark Lab
[3]Hong Kong University of Science and Technology (Guangzhou)
{jdtian,pchen,rhxu}@stu.ecnu.edu.cn, {yuxliang}@outlook.com
{cjguo,ayzhou,byang}@dase.ecnu.edu.cn, {raozhongwen,panlujia}@huawei.com

## ABSTRACT

Air pollution significantly threatens human health and ecosystems, necessitating effective air quality prediction to inform public policy. Traditional approaches are generally categorized into physics-based and data-driven models. Physics-based models usually struggle with high computational demands and closed-system assumptions, while data-driven models may overlook essential physical dynamics, confusing the capturing of spatiotemporal correlations. Although some physics-guided approaches combine the strengths of both models, they often face a mismatch between explicit physical equations and implicit learned representations. To address these challenges, we propose Air-DualODE, a novel physics-guided approach that integrates dual branches of Neural ODEs for air quality prediction. The first branch applies open-system physical equations to capture spatiotemporal dependencies for learning physics dynamics, while the second branch identifies the dependencies not addressed by the first in a fully data-driven way. These dual representations are temporally aligned and fused to enhance prediction accuracy. Our experimental results demonstrate that Air-DualODE achieves state-of-the-art performance in predicting pollutant concentrations across various spatial scales, thereby offering a promising solution for real-world air quality challenges. The code is available at: https://github.com/decisionintelligence/Air-DualODE.

## 1 INTRODUCTION

Air pollution presents a significant threat to both human health and the environment. For individuals, prolonged exposure elevates the risk of respiratory and cardiovascular diseases (Azimi & Rahman, 2024). From an environmental perspective, air pollution contributes to phenomena such as acid rain and ozone depletion, disrupting ecosystems and diminishing biodiversity (Cheng et al., 2024). These detrimental effects underscore the critical need for precise air quality forecasting, not only to protect public health but also to inform and shape governmental policies effectively.

Traditional research on air quality prediction largely relies on physics-based and data-driven approaches, each of which has their own inherent limitations. Physics-based models make predictions by solving Ordinary or Partial Differential Equations (ODEs/PDEs), which capture the transport of pollutants through diffusion and advection, as illustrated in Fig.1. Although these models achieve high accuracy, they require substantial computational resources, especially when applied on large spatial scales. Conversely, data-driven models excel at uncovering dependencies within data (Cheng et al., 2023; Xu et al., 2024a), yet their lack of integration with physical principles can lead to incomplete or even erroneous representations of spatiotemporal relationships. Therefore, it is crucial to propose a hybrid model that effectively harnesses the strengths of both physics-based and data-driven approaches. However, the development of such a model presents two primary challenges.

**Unrealistic assumptions of physical equations in an open air quality system.** Existing work employs fundamental continuity equations from physics to model the pollutant diffusion process (Mohammadshirazi et al., 2023; Hettige et al., 2024). These equations assume the region of interest is

---
*Corresponding author

(a) Diffusion: Pollutants diffuse from high concentration areas to low concentration areas.

(b) Advection: Pollutants spread to other areas under the influence of the wind field.

Figure 1: The heatmaps show $PM_{2.5}$ concentrations of individual stations at time steps $t$ and $t+1$. In this open system, there are sinks that absorbs pollutants and sources that generates new pollutants, along with pollutants exiting and entering the region's boundary through diffusion and advection.

a closed system, where the total mass remains constant over time. While such assumptions have proven effective in fields like traffic flow and epidemic modeling (Ji et al., 2022; Gao et al., 2021), they are insufficient for real-world air quality prediction, which operates in an open system where pollutants continuously enter and exit with airflow (e.g., A, D, E in Fig.1). Meanwhile, new pollutants may be introduced by industrial activity and vehicular emissions within the region (e.g., B, C, D, and E in Fig.1), while others are absorbed by natural elements like forests and lakes (e.g., A, F in Fig.1a). Given these facts, relying on closed-system assumptions fails to adequately model physical phenomena in open systems, and may introduce incorrect inductive biases into models.

**Mismatch between explicit physical equations and implicit deep learning representations.** Physical equations provide an explicit, interpretable framework for modeling pollutant concentration changes, with each variable tied to a well-defined physical meaning. In contrast, deep learning approaches capture spatiotemporal dependencies through implicit representations, often lacking direct physical interpretation. To harness the strengths of both methods, researchers have begun integrating physical equations into neural networks (Verma et al., 2024). These hybrid approaches design neural architectures around physical principles, ensuring that latent variables align with well-established physical meanings (Ji et al., 2022; Hettige et al., 2024; Nascimento et al., 2021). However, the high-dimensional latent spaces in neural networks often represent nonlinear combinations or abstract transformations of physical variables, making it inappropriate to interpret these dimensions as specific physical quantities. Therefore, one of the primary challenge in integrating physics into data-driven models is resolving this fundamental mismatch.

In this paper, we propose Air-DualODE, a Physics-guided Dual Neural ODEs that integrate physics dynamics and data-driven dynamics for open system's Air Pollution Prediction. For the first challenge, we propose a discrete Boundary-Aware Diffusion-Advection Equation (BA-DAE) to appropriately model open air systems. Due to the uncertainties associated with pollutant generation (source) and dissipation (sink) in open air systems, pollutant changes involve significant uncertainty, but both processes are linked to individual stations. To capture these dynamics, BA-DAE introduces a linear correction term, breaking away from the unrealistic closed system assumption and providing a more accurate description of physical phenomena in open air systems. Additionally, we incorporate Data-Driven Dynamics to capture spatiotemporal dependencies that BA-DAE ignores, making our approach better suited for real-world open air systems. For example, Data-Driven Dynamics can learn some patterns of pollutants generation, which cannot be described by BA-DAE.

To address the second challenge, we propose dual dynamics to integrate physical equations into data-driven models while retaining the model's ability to learn spatiotemporal correlations from data. Physics Dynamics generates spatiotemporal sequences consistent with physical phenomena by solving the BA-DAE, while Data-Driven Dynamics uses Neural ODE to learn dependencies not captured by physical equations. To place the physical knowledge and data-driven representations in the same space, we project the output of Physics Dynamics into latent spaces. However, these representations cannot be directly fused, as they correspond in time but are not fully matched. To resolve this, we design a dynamics fusion module with decaying temporal alignment to fuse both representations. In conclusion, we summarize our contributions as follows:

- Considering that air pollution transport is in an open system, we redefine the discrete diffusion-advection equation in explicit spaces and propose the BA-DAE, which makes physical equations more consistent with pollutant transport in open air systems.

- We introduce Air-DualODE, a model that integrates Physics Dynamics and Data-Driven Dynamics to harness the strengths of both physical knowledge and data-driven insights. To the best of

our knowledge, this is the first dual dynamics deep learning approach specifically designed for air quality prediction in open system.

- Experiments have shown that Air-DualODE achieves state-of-the-art results in predicting pollutant concentrations at both city and national spatial scales.

## 2 PRELIMINARIES

### 2.1 PROBLEM STATEMENT

Air quality prediction relies on historical data from $N$ observation stations, including pollutant concentrations $X_{1:T} \in \mathbb{R}^{T \times N \times 1}$, such as $PM_{2.5}$ and $NO_2$. It also incorporates auxiliary covariates $A_{1:T} \in \mathbb{R}^{T \times N \times (D-1)}$, like temperature and wind speed. The objective is to forecast future pollutant concentrations at these stations. Based on the region's geographical information, we construct a geospatial graph $G = (\mathbb{V}, \mathbb{E})$, where $\mathbb{V}$ is a set of stations $|\mathbb{V}| = N$, and $\mathbb{E}$ is a set of edges. Specifically, An edge $E_{ij}$ between nodes $V_i$ and $V_j$ indicates a potential pathway for pollutant transport (See Appendix A.4 for more details). The objective of the air quality prediction task is to learn a function $f(.)$ that can accurately forecast future pollutant concentrations $\hat{X}_{T+1:T+\tau} \in \mathbb{R}^{\tau \times N \times 1}$. This function can be formalized as:

$$\hat{X}_{T+1:T+\tau} = f(X_{1:T}, A_{1:T}, G).$$

### 2.2 BASIC CONCEPTS

**Continuity Equation.** The continuity equation, a fundamental principle in physics, describes mass transport. It explains that system mass changes over time due to material inflow and outflow, assuming no sources like vehicle emissions or sinks like forest absorption.

$$\frac{\partial X}{\partial t} + \vec{\nabla} \cdot (X\vec{F}) = 0, \tag{1}$$

where $X$ is air pollutant concentration, $\vec{F}$ is the flux of particles which describes the transport of pollutants particles, and $\vec{\nabla}\cdot$ is the divergence operator. When solving the Continuity Equation, it is crucial to determine if the environment is an open or closed system because the boundary conditions differ significantly, affecting the equation's discrete form.

**Diffusion-Advection Equation.** By modifying the flow field $\vec{F}$ of Eq.1, both the Diffusion and Advection equations for pollutants can be derived (Hettige et al., 2024). See Appendix A.2 for more details.

$$\frac{\partial X}{\partial t} = k \cdot \nabla^2 X - \vec{\nabla}(X \cdot \vec{v}), \tag{2}$$

where $k$ is diffusion coefficient, $\vec{v}$ is wind field, and $\nabla^2$ is a Laplacian operator combining divergence and gradient.

### 2.3 RELATED WORK

**Air Quality Prediction.** Air quality prediction is a key objective in smart city. Existing approaches can be broadly categorized into two types: physics-based models and deep learning models. The former rely on domain knowledge to construct Ordinary and Partial Differential Equations (ODEs/PDEs) that describe air pollutant transport (Li et al., 2023; Daly & Zannetti, 2007). However, these models face two major limitations. They rely on closed-system assumptions, and their use of numerical methods, such as finite difference or finite element methods (Özişik et al., 2017; Ŝolín, 2005), becomes computationally expensive at larger spatial scales. This makes rapid prediction difficult for urban planning (Lai et al., 2023) and policy-making (Liu et al., 2025). Deep learning (Campos et al., 2023; 2024; Qiu et al., 2024; Chen et al., 2024; Qiu et al., 2025) methods have recently gained prominence by using historical data and auxiliary covariates, such as weather, to extract high-dimensional spatiotemporal representations for pollutant prediction. These models typically employ RNNs or Transformers for temporal representations and GNNs or CNNs for spatial dependencies (Yu et al., 2017; Shang et al., 2021; Zhao et al., 2023; Wu et al., 2024). Furthermore,

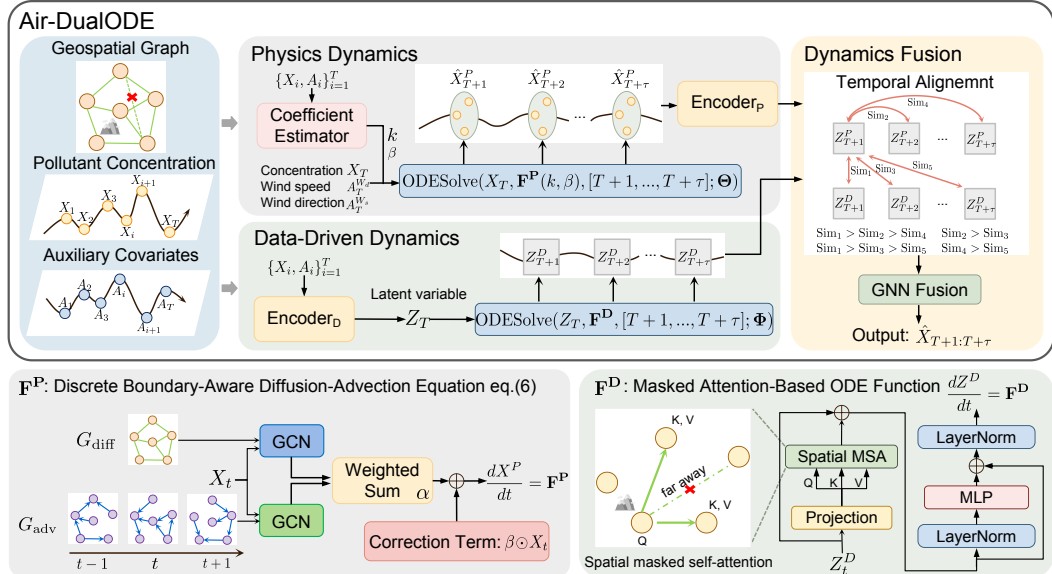

Figure 2: The overall framework of Air-DualODE consists of Physics Dynamics, Data-Driven Dynamics, and Dynamics Fusion. $\mathbf{F^P}$ and $\mathbf{F^D}$ represent the ODE functions in Physics Dynamics and Data-Driven Dynamics, respectively.

there are models specifically designed for air pollutant prediction tasks (Yu et al., 2023; Liang et al., 2023). However, these deep learning approaches may overlook essential physical dynamics, leading to confused spatiotemporal correlations that could violate physical principles.

**Physics Guided Deep Learning.** Recently, several studies have explored incorporating physical knowledge into deep learning models (Karpatne et al., 2024). The most direct approach involves using physical knowledge as constraints in loss functions (Raissi et al., 2017; Shi et al., 2021), penalizing deviations from physical laws to encourage physically consistent patterns and improve generalizability (Greydanus et al., 2019; Choi et al., 2023b). However, these methods rely on highly accurate physical knowledge; otherwise, they may introduce inductive bias, limiting the model's learning capacity. Another approach is to design hybrid frameworks that integrate scientific knowledge into specific modeling components (Bao et al., 2021; Xu et al., 2024b), such as incorporating physical equations into Neural ODEs (Hwang et al., 2021; Ji et al., 2022; Choi et al., 2023a; Hettige et al., 2024). These methods often require deep latent representations to align with physical meanings, forcing models to switch between abstract representations and physical quantities Nascimento et al. (2021); Verma et al. (2024).

## 3 METHODOLOGY

### 3.1 MODEL OVERVIEW

To integrate physical knowledge into neural networks, and ensure accurate spatiotemporal correlations, we propose Air-DualODE, which includes BA-DAE and dual dynamics under open air systems. As shown in Fig.2, the framework consists of three main components:

- **Physics Dynamics** To maintain consistency with the neural network's implicit representations, Physics Dynamics directly solves physical equations to obtain spatiotemporal correlations with physical meanings. Specifically, it solves the BA-DAE, namely $\mathbf{F^P}$, a more realistic equation for open air systems, to generate physical simulation results $\hat{X}_{T+1:T+\tau}^P$ for the next $\tau$ steps. These results are then mapped into the latent space as $Z_{T+1:T+\tau}^P$.

- **Data-Driven Dynamics** Although BA-DAE captures pollutant changes in an open air system, it does not account for some spatiotemporal dependencies, like the effects of temperature and humid-

ity on pollutant transport. To address this, Data-Driven Dynamics is implemented using a Neural ODE with spatial masked self-attention (Spatial-MSA) in ODE function $\mathbf{F^D}$. This branch captures unknown dynamics in latent space and generates latent dynamics representation $Z^D_{T+1:T+\tau}$.

- **Dynamics Fusion** Although $Z^P_{T+1:T+\tau}$ and $Z^D_{T+1:T+\tau}$ share the same latent space and time span, they are not yet aligned. To resolve this, Decaying Temporal Contrastive Learning (Decay-TCL) aligns them over time using decaying weights for effective fusion. A GNN then spatially fuses two representations, and the combined result is decoded to generate the prediction $\hat{X}_{T+1:T+\tau}$.

### 3.2 Physical dynamics of Open System's Air Pollution

#### 3.2.1 Discrete Diffusion-Advection Equation under closed system

The Method of Lines (MOL) can discretize the PDEs (Eq.2) into a grid of location-specific ODEs (Schiesser, 2012; Iakovlev et al., 2021). The discrete equation can be derived as follows:

$$\frac{dX}{dt} = -k \cdot L_{\text{diff}}X + L_{\text{adv}}X. \tag{3}$$

$L_{\text{diff}}$ and $L_{\text{adv}}$ are the discretized graph Laplacian operators for diffusion and advection, respectively, approximated using the Chebyshev GNN as referenced in (Defferrard et al., 2016; Hettige et al., 2024). According to Chapman & Chapman (2015), it is necessary to define distinct graph structures to represent diffusion on concentration gradient fields and advection on wind fields.

**Diffusion Graph** is an undirected weighted graph based on $G$, with weights computed using the inverse of the distance between nodes, such that $G_{\text{diff}}[ij] = \frac{1}{d_{ij}}$ (Han et al., 2023), where $d_{ij}$ represents the haversine distance between nodes $V_i$ and $V_j$.

**Advection Graph** is a directed weighted graph based on $G$, where the direction and weight of the edges depend on the wind direction $A^{W^d}_T$, and wind speed $A^{W^s}_T$ at $T$ timestamps. After calculating the included angle $\xi$ in Fig.3, we project the wind vector of $V_i$ onto $E_{ij}$ to obtain the wind intensity from $V_i$ to $V_j$, as shown in the Fig.3. The $G_{\text{adv}}[ij]$ is calculated as follows:

$$G_{\text{adv}}[ij] = \begin{cases} \text{ReLU}\left(\frac{|v|}{d_{ij}} \cdot \cos\xi\right) & \text{when } G_{ij} = 1, \\ 0 & \text{when } G_{ij} = 0. \end{cases} \tag{4}$$

In particular, if the wind vector's projection at $V_i$ does not point in the direction of $V_j$, i.e., $\cos\xi < 0$, this suggests that the pollutants at $V_i$ are currently not influenced by advection to reach $V_j$. Consequently, at that time, the directed edge in the advection graph will not be constructed. Notably, $G_{\text{adv}}$ differs from $G_{\text{diff}}$. $G_{\text{diff}}$ remains static over time, representing a static graph. In contrast, $G_{\text{adv}}$ changes dynamically over time as the wind speed and direction at each node vary, making it a dynamic graph that varies at different timestamps.

It is easy to prove that Eq.3 has conservation property (see Appendix A.3), which means the amount of pollutants concentration remaining constant over time:

$$\sum_{i=1}^{N} \frac{dX_i}{dt} = 0. \tag{5}$$

#### 3.2.2 Discrete Boundary-Aware Diffusion-Advection Equation for open system

Since air pollutant propagation does not occur in a closed system, the total pollutant mass cannot remain constant over time. Particularly, pollutants can dissipate beyond boundaries, such as being carried away by wind in Fig.1b. Additionally, industrial zones and vehicles contribute to pollution, while urban lakes and rainfall absorb pollutants. Thus, the dynamics of air pollution are far more complex than a closed system's conservation assumption.

In an open system, changes in the total amount of pollutants can be categorized into dissipation and generation. Pollutant dissipation refers to sinks, including transport out of the boundary through diffusion and advection, as well as absorption by forests or lakes. Pollutant generation refers to sources such as industrial emissions. As shown in Fig.4, the dissipation and generation at Point E

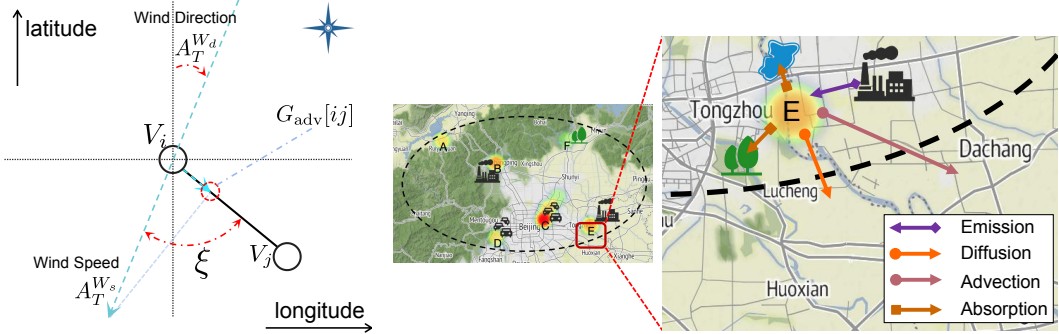

Figure 3: Construction of $G_{\mathrm{adv}}[ij]$.      Figure 4: The sources and sinks at station E.

are highly relevant to its current concentrations. Therefore, we incorporate two terms into Eq.3: the dissipation term $\boldsymbol{\beta}^- \odot X$ ($\boldsymbol{\beta}^- \in [-1, 0]^{N \times 1}$) and the generation term $\boldsymbol{\beta}^+ \odot X$ ($\boldsymbol{\beta}^+ \in [0, +\infty)^{N \times 1}$). These terms can be simplified into an open system correction term: $\boldsymbol{\beta} \odot X$ ($\boldsymbol{\beta} \in [-1, +\infty)^{N \times 1}$). Boundary stations, experiencing more outward diffusion and advection, tend to have greater negative $\beta_i$. So, this equation is called as Discrete Boundary-Aware Diffusion-Advection Equation (BA-DAE):

$$\mathbf{F}^{\mathbf{P}}(X_t; \boldsymbol{\Theta}) = \frac{dX}{dt} = \boldsymbol{\alpha} \odot (-k \cdot L_{\mathrm{diff}} X) + (1 - \boldsymbol{\alpha}) \odot (L_{\mathrm{adv}} X) + \boldsymbol{\beta} \odot X. \quad (6)$$

Here, $\boldsymbol{\Theta}$ represents learnable parameters used to approximate the graph Laplacian operators. The gate value $\boldsymbol{\alpha} \in \mathbb{R}^{N \times 1}$ is estimated by a linear layer ($\mathbb{R}^2 \to \mathbb{R}$) based on each station's diffusion and advection over time, representing the weighted influence of these two effects as they vary across time and space. The diffusion coefficient $k$ and the correction term's coefficient $\boldsymbol{\beta}$ are estimated via the Coefficient Estimator, a Recurrent Neural Network. By solving BA-DAE with ODE solver from Chen et al. (2018), pollutant concentrations from $T + 1$ to $T + \tau$ are obtained, and a temporal encoder maps these concentrations to the latent space for dynamics fusion, as illustrated in the Physics Dynamics part of Fig 2.

$$\hat{X}^P_{T+1:T+\tau} = \mathrm{ODESolve}(X_T, \mathbf{F}^{\mathbf{P}}, [T + 1, ..., T + \tau]; \boldsymbol{\Theta}), \quad (7)$$

$$Z^P_{T+1:T+\tau} = \mathrm{Encoder}_P(\hat{X}^P_{T+1:T+\tau}), \quad (8)$$

where $Z^P_{T+1:T+\tau}$ encodes the spatiotemporal dependencies of pollutant transport with physical knowledge, which data-driven models may not fully capture. This compensates for the limitations of data-driven approaches and enhances the interpretability of the model.

### 3.3 DATA-DRIVEN NEURAL ODE

The Data-Driven Neural ODE focuses on modeling dynamical systems that go beyond the limitations of the BA-DAE, aiming to uncover knowledge that cannot be described by physical equations alone. For instance, spatiotemporal correlations within historical data patterns are not adequately described by the BA-DAE. However, data-driven dynamics can bridge these gaps by learning from historical data. Additionally, relationships such as humidity and pollutant transport can also be effectively captured through this branch. Following Chen et al. (2018), Latent-ODE is employed in the Data-Driven Dynamics branch, as depicted in Fig.2. Specifically, we utilize $(X, A)_{1:T}$ as input, which is mapped to the latent variables $Z^D_T$ via an Encoder$_D$. This serves as the initial condition for learning and solving the Neural ODE, resulting in $Z^D_{T+1:T+\tau}$.

$$Z^D_T = \mathrm{Encoder}_D(X_{1:T}, A_{1:T}), \quad (9)$$

$$Z^D_{T+1:T+\tau} = \mathrm{ODESolve}(Z^D_T, \mathbf{F}^{\mathbf{D}}, [T + 1, ..., T + \tau]; \boldsymbol{\Phi}), \quad (10)$$

where $\boldsymbol{\Phi}$ represents the learnable parameters in ODE function $\mathbf{F}^{\mathbf{D}}$. To capture spatial correlations across different time steps within the dynamic system, a masked self-attention mechanism is incorporated into $\mathbf{F}^{\mathbf{D}}$. Specifically, the adjacency matrix of $G$ is employed as a mask in the Spatial-MSA since intuitively, if no potential transport pathway exists between two stations, their representations should not be correlated. This approach not only improves computational efficiency but also enables the model to focus on relevant information, enhancing its effectiveness across various spatial granularities in prediction scenarios.

### 3.4 DYNAMICS FUSION

**Decaying Temporal Constrastive Learning.** Before combining the physics and data-driven dynamics, it is essential to align their latent representations, $Z_{T+1:T+\tau}^P$ and $Z_{T+1:T+\tau}^D$, to resolve inconsistencies and enable effective fusion. Since both dynamics operate synchronously, three key intuitions are considered: 1) latent representations at the same time step are more similar than those at different time steps, 2) adjacent time steps are more similar than distant ones, and 3) representations from the same dynamics are more similar than those from different dynamics. To satisfy the three intuitions, we propose Decaying Temporal Contrastive Learning (Decay-TCL). It computes a decaying weight $w(t, s)$ based on timestamp differences and dynamics type to regulate the similarity between latent representations. The hyperparameters $\lambda_1$ and $\lambda_2$ are used to differentiate the distinct dynamics, and they must satisfy $\lambda_1 > \lambda_2$ to fulfill third intuition.

$$w(t,s) = \begin{cases} 2 \cdot \sigma(-\lambda_1 \cdot |t-s|) & \text{if } |t-s| < \tau \\ 2 \cdot \sigma(-\lambda_2 \cdot |t-s \bmod \tau|) & \text{if } |t-s| > \tau, \end{cases}$$

where $\sigma(\cdot)$ represents sigmoid function. Let $\bar{\mathcal{Z}}_{1:3\tau} = [Z^P, Z^D, Z^P]$. We define the softmax probability of the relative similarities when computing the loss as:

$$p(t, t') = \frac{\exp\{\text{Sim}(\bar{\mathcal{Z}}_t, \bar{\mathcal{Z}}_{t'})\}}{\sum_{i=1, i \neq t'}^{2\tau} \exp\{\text{Sim}(\bar{\mathcal{Z}}_t, \bar{\mathcal{Z}}_i)\}}.$$

The function $\text{Sim}(\cdot, \cdot)$ denotes cosine similarity. The decaying temporal contrastive loss for $\bar{\mathcal{Z}}$ at timestamp $t$ is defined as:

$$\ell(t) = -\log p(t, t+\tau) - \sum_{j=1, j \neq \{t, t+\tau\}}^{2\tau} w(t, j) \cdot \log p(t, j). \tag{11}$$

Eq.11 ensures the similarity inequality in the Temporal Alignment shown in Fig.2, aligning the physical and data-driven representations in the temporal dimension to enable better fusion. The final loss for Decay-TCL is defined as:

$$\mathcal{L}_{\text{tcl}} = \frac{1}{2\tau} \sum_{t=1}^{2\tau} \ell(t). \tag{12}$$

**GNN Fusion.** After temporal alignment, the two latent representations are concatenated as $Z_t^{(0)} = \text{concat}(Z_t^P, Z_t^D)$ at each timestamp. Since the influence of stations is distance-dependent, using MLP structures may introduce noises from distant sites, complicating the learning process thus reducing accuracy (see Appendix A.16). To address this, GNNs are employed based on the geospatial graph $G$ to Dynamics Fusion: $Z_t^{(n)} = \text{GNNs}(Z_t^{(0)})$, where $n$ represents the GNN layers. The Decoder then transforms the latent representations $Z_{T+1:T+\tau}^{(n)}$ into predicted pollutant concentrations $\hat{X}_{T+1:T+\tau}$. We jointly optimize the prediction loss and the loss functions in Eq.12, with a hyperparameter $\gamma$ to balance them:

$$\mathcal{L} = \frac{1}{\tau} \sum_{t=T+1}^{T+\tau} \|X_t - \hat{X}_t\| + \gamma \mathcal{L}_{\text{tcl}}. \tag{13}$$

## 4 EXPERIMENT

### 4.1 EXPERIMENT SETTING

**Datasets.** We evaluate the performance of our model using two real-world air quality datasets: the Beijing[1] dataset and the KnowAir[2] dataset. The Beijing dataset is evaluated at the city scale, while the KnowAir dataset is evaluated at the national scale. Further descriptions about these two datasets can be found in Appendix A.5. In accordance with previous studies (Liang et al., 2023), we focus on $PM_{2.5}$ concentration as the target variable, using meteorological factors—including temperature,

---

[1] https://www.biendata.xyz/competition/kdd_2018/
[2] https://github.com/shuowang-ai/PM2.5-GNN

pressure, humidity, wind speed, and direction—as auxiliary covariates to ensure consistency between the two datasets. Similar to prior work (Liang et al., 2023; Hettige et al., 2024), we employ a 3-hour time step and utilize data from the preceding 72 hours (24 steps) to predict the subsequent 72 hours. More details regarding the implementation can be found in Appendix A.7. Additionally, following Liang et al. (2023), we analyze the errors in predicting sudden changes. Sudden changes are defined as cases where $PM_{2.5}$ levels exceed 75 μg/m³ and fluctuate by more than $\pm 20$ μg/m³ in the next three hours.

**Baselines.** We evaluate Air-DualODE against baselines of four categories: 1) Classical Methods: Historical Average (HA) and Vector Auto-Regression (VAR). 2) Differential Equation Network-Based Methods: Latent-ODE (Chen et al., 2018), ODE-RNN (Rubanova et al., 2019) and ODE-LSTM (Lechner & Hasani, 2020). 3) Spatio-temporal Deep Learning Methods: DCRNN (Li et al., 2017), STGCN (Yu et al., 2017), ASTGCN (Guo et al., 2019), GTS (Shang et al., 2021), MTSF-DG (Zhao et al., 2023), PM2.5-GNN (Wang et al., 2020), and AirFormer (Liang et al., 2023). 4) Physics-guided Neural Networks: AirPhyNet (Hettige et al., 2024).

**Evaluation Metrics.** Baselines are evaluated using Mean Absolute Error (MAE), Root Mean Square Error (RMSE), and Symmetric Mean Absolute Percentage Error (SMAPE), with smaller values indicating better performance (see in Appendix.A.6).

## 4.2 PERFORMANCE COMPARISON

**Overall Performance.** Table 1 compares the performance of Air-DualODE with baselines over three days and sudden changes across two datasets. Air-DualODE outperforms all baselines across all metrics for both datasets. These results demonstrate the model's effectiveness in integrating Physics Dynamics and Data-Driven Dynamics within a deep learning framework to capture complex spatiotemporal dynamics relationships.

Table 1: Overall prediction performance comparison. **Bold** fonts indicate the best results, while underlined fonts signify the second-best results.

| Methods | Beijing | | | | | | KnowAir | | | | | |
| | 3days | | | Sudden Change | | | 3days | | | Sudden Change | | |
| | MAE | RMSE | SMAPE | MAE | RMSE | SMAPE | MAE | RMSE | SMAPE | MAE | RMSE | SMAPE |
|---|---|---|---|---|---|---|---|---|---|---|---|---|
| HA | 59.78 | 77.73 | 0.83 | 79.66 | 94.87 | 0.87 | 25.27 | 38.57 | 0.53 | 59.37 | 79.38 | 0.72 |
| VAR | 55.88 | 75.64 | 0.82 | 77.18 | 98.73 | 0.86 | 24.56 | 37.35 | 0.51 | 57.40 | 71.84 | 0.71 |
| LatentODE | 43.35 | 63.75 | 0.79 | 69.71 | 92.67 | 0.77 | 19.99 | 30.85 | 0.46 | 42.59 | 57.90 | 0.50 |
| ODE-RNN | 43.60 | 63.67 | 0.79 | 70.22 | 93.12 | 0.78 | 20.57 | 31.26 | 0.45 | 42.07 | 57.76 | 0.51 |
| ODE-LSTM | 43.68 | 64.23 | 0.78 | 70.60 | 93.27 | 0.77 | 20.21 | 30.88 | 0.45 | 41.29 | 56.47 | 0.49 |
| STGCN | 45.99 | 64.85 | 0.82 | 73.65 | 97.87 | 0.76 | 23.64 | 32.48 | 0.52 | 55.29 | 73.21 | 0.54 |
| DCRNN | 49.61 | 71.53 | 0.82 | 75.31 | 96.50 | 0.81 | 24.02 | 37.87 | 0.53 | 56.88 | 74.58 | 0.73 |
| ASTGCN | 42.88 | 65.24 | 0.79 | 72.75 | 96.68 | 0.83 | 19.92 | 31.39 | 0.44 | 42.06 | 57.50 | 0.52 |
| GTS | 42.96 | 63.27 | 0.80 | 70.36 | 92.70 | 0.76 | 19.52 | 30.36 | 0.43 | 40.87 | 56.20 | 0.49 |
| MTSF-DG | 43.12 | 66.06 | 0.79 | 72.26 | 95.32 | 0.77 | 19.17 | 29.55 | 0.43 | 40.64 | 55.65 | 0.50 |
| PM25GNN | 45.23 | 66.43 | 0.82 | 72.45 | 95.34 | 0.78 | 19.32 | 30.12 | 0.43 | 40.43 | 55.49 | 0.49 |
| Airformer | 42.74 | 63.11 | 0.78 | 68.80 | 91.16 | 0.75 | 19.17 | 30.19 | 0.43 | 39.99 | 55.35 | 0.49 |
| AirPhyNet | 42.72 | 64.58 | 0.78 | 70.03 | 94.60 | 0.78 | 21.31 | 31.77 | 0.47 | 43.23 | 58.79 | 0.50 |
| Air-DualODE | **40.32** | **62.04** | **0.74** | **66.40** | **90.31** | **0.73** | **18.64** | **29.37** | **0.42** | **39.79** | **54.61** | **0.49** |

From Table 1 we can also observe the following: 1) deep learning approaches outperform classical methods such as HA and VAR, demonstrating their superior ability to capture complex spatiotemporal correlations. 2) Spatiotemporal deep learning models, originally developed for traffic flow forecasting, such as ASTGCN and GTS, exhibit strong adaptability to air quality forecasting, producing results comparable to SOTA air quality prediction methods like AirFormer and PM25GNN. 3) Approaches based on Neural DE networks also exhibit competitive performance compared to spatiotemporal deep learning methods, as they capture the ODE function in latent space, which is crucial for accurate air quality prediction. Overall, these results suggest the effectiveness of Air-DualODE in achieving precise air quality predictions while leveraging the physics-based domain knowledge of air pollutant transport. More Discussions about sudden change evaluation can be found in Appendix A.17.

### 4.3 ABLATION STUDY

**Effect of Dual Dynamics.** To validate the effectiveness of Dual Dynamics, we test each branch of Air-DualODE independently across both datasets: a) **w/o Physics Dynamics**, which solely relies on Data-Driven Dynamics augmented by the spatial-MSA mechanism; b) **w/o Data-Driven Dynamics**, employing only Physics Dynamics without the encoder. Both configurations exclude Dynamics Fusion. As shown in Table 2, the removal of either branch results in diminished performance. Notably, the absence of Data-Driven Dynamics leads to notably poor performance from BA-DAE on both datasets, since physical equations often idealize real-world scenarios. Conversely, the integration of both branches significantly enhances performance, demonstrating that Data-Driven and Physics Dynamics provide complementary insights.

Table 2: Effect of dual dynamics, fusion on latent space, Decay-TCL and Spatial-MSA.

| Methods | Beijing | | | KnowAir | | |
|---|---|---|---|---|---|---|
| | MAE | RMSE | SMAPE | MAE | RMSE | SMAPE |
| w/o Physics Dynamics | 42.75 | 63.45 | 0.75 | 19.69 | 30.65 | 0.45 |
| w/o Data-Driven Dynamics | 44.33 | 65.60 | 0.82 | 21.21 | 33.09 | 0.56 |
| Explicit Fusion | 41.32 | 63.08 | 0.76 | 19.06 | 30.28 | 0.43 |
| Cross-Space Fusion | 41.97 | 63.12 | 0.77 | 19.27 | 30.54 | 0.44 |
| w/o Decay-TCL | 42.34 | 66.56 | 0.82 | 19.10 | 29.79 | 0.43 |
| w/o Spatial-MSA | 40.52 | 62.49 | 0.75 | 18.97 | 30.01 | 0.43 |
| Air-DualODE | **40.32** | **62.04** | **0.74** | **18.64** | **29.37** | **0.42** |

**Effect of Fusion on Latent Space.** To further validate Dynamics Fusion in the latent space, we test alternative fusion strategies: a) **Explicit Fusion**: After solving ODE-Func $F_D$ in the latent space using Data-Driven Dynamics, we apply a Decoder to map it to explicit space (i.e., pollutant concentration) before fusion. b) **Cross-Space Fusion**: We combine the explicit-space output of Physics Dynamics with the deep representations of Data-Driven Dynamics. Both methods use GNN for fusion, consistent with Air-DualODE. As shown in Table 2, both methods led to a decline in performance. Fusion in explicit space experienced information loss during the decoding of deep representations, whereas fusion across different spaces encounter mismatched representations, which exacerbats performance deterioration. This underscores the importance of aligning and fusing Dynamics within the latent space to achieve optimal results.

**Effect of Physical knowledge.** To further demonstrate the impact of incorporating physical knowledge, we compare Air-DualODE with three other closed-system variations: a) **Diff-c**, which only considers the diffusion process, b) **Adv-c**, which only considers the advection process, and c) **DAE-c**, which accounts for both processes. Additionally, we compare these with d) **BA-DAE-o**, namely Air-DualODE, as proposed in Section 3.2.2 for open air systems. Fig.5 shows that integrated physical knowledge is crucial, and inadequate modeling leads to reduced performance. Consistent with Air-

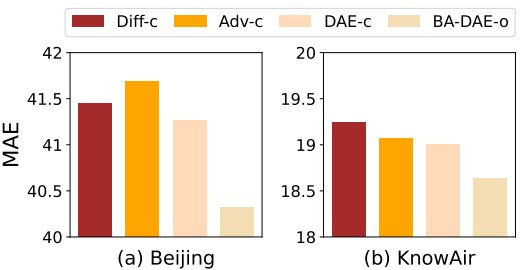

Figure 5: Effect of Physical Knowledge on MAE.

PhyNet, DAE-c outperforms Diff-c and Adv-c, but its inability to model dissipation and generation makes it less effective than BA-DAE-o. This underscores the effectiveness of BA-DAE in modeling open air systems. While Data-Driven Dynamics can compensate for some deficiencies in the physical equations, it still cannot match the effectiveness of direct physical modeling.

**Effect of Spatial-MSA.** To assess the impact of Spatial-MSA in $F^D$, we compare it with **w/o Spatial-MSA**, which uses standard self-attention (SA). Table 2 suggests that the performance without Spatial-MSA is similar to Air-DualODE on the Beijing dataset, while there is a significant performance gap on the KnowAir dataset. This is because stations in Beijing are geographically close with no major barriers, making Spatial-MSA almost equivalent to SA. In contrast, with KnowAir being a national dataset with distant stations, the performance gap highlights that distant station representations offer little useful information.

**Effect of Decay-TCL.** To validate Decay-TCL, we test the model **w/o Decay-TCL**, where $\mathcal{L}_{\text{tcl}}$ is removed from Air-DualODE. Table 2 shows that directly fusing Dual Dynamics in the latent space leads to a performance drop. This is because the latent representations of Physics Dynamics, after being mapped through the Encoder, may not align with those of Data-Driven Dynamics, resulting in mismatches and inconsistencies during fusion.

## 4.4 CASE STUDY

We present two case studies to demonstrate the interpretability of the physical equations within the dual dynamics. To highlight its physical significance, we visualize the predicted $PM_{2.5}$ concentrations alongside wind direction data from the Beijing dataset. Fig.6 displays the PM levels at monitoring stations at two consecutive prediction timestamps. The emergence of a southeast wind in Fig.6 causes the pollutant concentrations to shift in the direction of the wind, most noticeably in the circled regions. Specifically, $PM_{2.5}$ concentrations decrease in the purple-circled regions, while they increase in the black-circled regions. This indicates that Air-DualODE successfully captures certain spatiotemporal correlations between wind direction and pollutant concentrations based on the Advection equation in the BA-DAE.

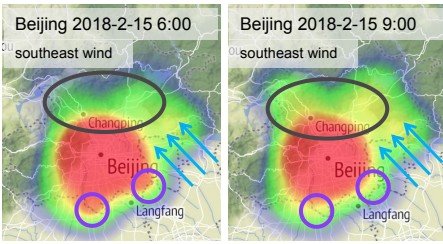 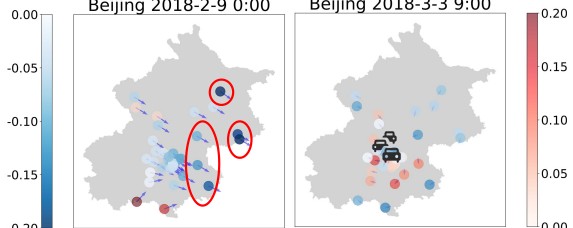

Figure 6: Visualization of predicted $PM_{2.5}$ concentrations under advection. The heatmap shows $PM_{2.5}$ concentration, while arrows represent wind direction.

Figure 7: Visualization of $\boldsymbol{\beta}$, where negative and positive values separately indicate dissipation and generation. The arrow direction represents wind direction, and the length reflects wind speed at each station.

Next, we visualize the $\boldsymbol{\beta}$ in BA-DAE under two scenarios on the Beijing dataset (see Fig.7). As described in Section 3.2.2, negative values represent $PM_{2.5}$ dissipation (blue colorbar), while positive values indicate $PM_{2.5}$ generation (red colorbar). We observe two distinct scenarios: In the first scenario (left side of Fig.7), with wind present, dissipation significantly increases at boundary stations in the wind direction (denoted by circles), resulting in larger negative $\beta_i$. Additionally, some boundary stations exhibit positive $\beta_i$, which is reasonable because the wind at these stations carries pollutants from outside the region. In other words, this can be considered as pollutant generation. In the second scenario (right side of Fig.7), with no wind, boundary stations exhibit greater dissipation due to forest absorption and diffusion beyond the boundary, leading to negative $\beta_i$ at the outskirts of Beijing. Notably, the city center of Beijing exhibits higher $\beta_i$ at 9 a.m., indicating pollutant generation. This is expected, as 9 a.m. coincides with Beijing's morning rush hour, when a large number of vehicles are present in the area. This example demonstrates that our model successfully captures the increase in pollutants caused by vehicle emissions. Overall, by integrating physical equations, Air-DualODE demonstrates excellent interpretability.

## 5 CONCLUSION AND FUTURE WORKS

We design Air-DualODE to predict air quality at both city and national levels. Our method employs a dual-dynamics approach, leveraging Neural ODEs to model physical (known) and data-driven (unknown) dynamics, which are subsequently aligned and fused in the latent space. By integrating physical equations into deep learning, Air-DualODE allows existing physical knowledge to enhance the model's learning, inference, and ultimately, its prediction accuracy. While the results are promising, predicting sudden changes remains a significant challenge in air quality forecasting. Therefore, Designing specific mechanisms to better handle sudden changes is an important and promising research direction. Moreover, we plan to further explore Physics-guided DualODE and extend its application to more general spatiotemporal prediction tasks.

ACKNOWLEDGEMENT

This work is partially supported by National Natural Science Foundation of China No.62372179. Yuxuan Liang's work is supported by the National Natural Science Foundation of China No.62402414.

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

# A  APPENDIX

## A.1  NOTATION

Table 3: Summary of Notations used in the paper

| Symbol | Description |
|:---:|:---|
| $G$ | Geospatial Graph |
| $G_{\text{diff}}$ | Diffusion Graph (static) |
| $G_{\text{adv}}$ | Advection Graph (dynamic) |
| $N$ | Number of Nodes |
| $\mathbb{V}, V_i$ | Nodes of the $G$ and $i$-th node |
| $\mathbb{E}$ | Set of Edges in $G$ |
| $X_t$ | Pollutant Concentration at $t$ timestamp ($X_t \in \mathbb{R}^{N \times 1}$) |
| $A_t$ | Auxiliary covariates at $t$ timestamp ($A_t \in \mathbb{R}^{N \times (D-1)}$) |
| $\vec{F}$ | Flux of Particles |
| $\vec{v}$ | Wind Field |
| $L_{\text{diff}}$ | Discrete Laplacian operator on $G_{\text{diff}}$ |
| $L_{\text{adv}}$ | Discrete Laplacian operator on $G_{\text{adv}}$ |
| $A_t^{W_s}$ | Wind speed at at $t$ timestamp |
| $A_t^{W_d}$ | Wind direction at at $t$ timestamp |
| $k$ | Diffusion Coefficient |
| $\boldsymbol{\beta}$ | Correction term's coefficient in BA-DAE ($\boldsymbol{\beta} \in R^{N \times 1}$) |
| $\beta_i$ | $\boldsymbol{\beta} = [\beta_1, ..., \beta_i, ..., \beta_N]$ |

## A.2  THE DERIVATION OF DIFFUSION-ADVECTION EQUATION

According to the Eq.1, by modifying the flow field $\vec{F}$, both the Diffusion and Advection equations for pollutants can be derived. $X(p, t)$ denotes the spatiotemporal distribution of pollutants, where $p$ corresponds to the spatial variable and $t$ refers to the temporal variable.

**Diffusion Equation** The driving flow field for Diffusion is dictated by the distribution of pollutant concentrations, specifically the gradient field of the pollutants. Thus, the Diffusion equation describes the phenomenon where pollutants always diffuse from areas of higher concentration to areas of lower concentration.

$$\frac{\partial X}{\partial t} - k \cdot \nabla^2 X = 0. \tag{14}$$

**Advection Equation** The driving flow field for Advection is the natural wind field. Consequently, the Advection equation describes the phenomenon where pollutants propagate in the direction of the wind, with the magnitude of propagation dependent on the wind speed.

$$\frac{\partial X}{\partial t} + \vec{\nabla}(X \cdot \vec{v}) = 0. \tag{15}$$

The Method of Lines (MOL) can discretize the PDEs into a grid of location-specific ODEs for solving Eq.14 and Eq.15 (Schiesser, 2012). According to Chapman & Chapman (2015), the discrete equation can be drived as follow:

**Discrete Diffusion Equation**

$$\frac{dX_i}{dt} = k \cdot \sum_{j \in N(i)} G_{\text{diff}}[ij](X_i - X_j), \tag{16}$$

where $G_{\text{diff}}[ij]$ represents the pollutants transport weight from $V_i$ to $V_j$ under the diffusion scenario, $k$ denotes diffusion coefficient, and $N(i)$ denotes the set of neighbours of $V_i$.

**Discrete Advection Equation**

$$\frac{dX_i}{dt} = -\sum_{\forall k \leftarrow i} G_{\text{adv}}[ik]X_i + \sum_{\forall j \rightarrow i} G_{\text{adv}}[ji]X_j, \tag{17}$$

where $G_{\text{adv}}[ij]$ represents the transport of pollutants from $V_i$ to $V_j$ under the advection scenario. The first term indicates that the pollutant concentration at $V_i$ decreases due to the wind blowing pollutants from $V_i$ to $V_j$, while the second term reflects the contribution of pollutants transported by the wind from other stations to $V_i$.

A.3 PROOF OF CLOSED SYSTEM'S CONSERVATION PROPERTY

In this section, we use the Discrete Diffusion Equation (Eq.16) as an example to illustrate the mass conservation law in a closed system. By applying the Method of Lines (MOL) for spatial discretization, we derive the spatially discrete form of Eq.14. Next, we demonstrate the validity of Eq. 5.

$$\begin{cases} \dfrac{dX_i}{dt} = -k \cdot L_{\text{diff}}X = k \cdot \sum_{j \in N(i)} G_{\text{diff}}[ij](X_i - X_j), \quad \forall i \in N \\[2em] \text{Given} \quad X_i(0). \end{cases}$$

Proof:

$$\sum_{i=1}^{N} \frac{dX_i}{dt} = k \cdot \sum_{i=1}^{N} \sum_{j \in N(i)} G_{\text{diff}}[ij](X_i - X_j)$$

$$= k \cdot \sum_{i=1}^{N} \left[ \sum_{j \in N(i)} G_{\text{diff}}[ij]X_i - \sum_{j \in N(i)} G_{\text{diff}}[ij]X_j \right]$$

$$= k \cdot \sum_{i=1}^{N} \left[ X_i \sum_{j \in N(i)} G_{\text{diff}}[ij] - \sum_{j \in N(i)} G_{\text{diff}}[ij]X_j \right]$$

$$= k \cdot \sum_{i=1}^{N} X_i d_i - k \cdot \sum_{i=1}^{N} \sum_{j \in N(i)} G_{\text{diff}}[ij]X_j$$

$$= 0$$

The final equality holds because the number of times $X_j$ is added depends on how many vertices it is connected to, which is represented by its degree, $d_i$.

It follows the same principle as the discrete Advection equation, meaning the discrete Diffusion-Advection Equation also adheres to the mass conservation law.

A.4 GEOSPATIAL GRAPH

It is essential to clarify the conditions under which pollutants can propagate. According to the settings in Wang et al. (2020), pollutants cannot propagate when the distance is too great or when geographical barriers such as mountains exist between stations. The formula is as follows:

$$G_{\text{Geo-Spatial}}[ij] = \text{H}(d_\theta - d_{ij}) \cdot \text{H}(m_\theta - m_{ij})$$

$$\begin{cases} d_{ij} = \|\rho_i - \rho_j\| \\[1em] m_{ij} = \sup_{\lambda \in (0,1)} \{h[\lambda \rho_i + (1-\lambda)\rho_i] - \text{Max}\{h(\rho_i), h(\rho_j)\}\} \end{cases}$$

In this context, $H(\cdot)$ denotes the Heaviside step function, which outputs 1 if the value is greater than 0 and 0 if it is less than 0. The symbol $h(\cdot)$ represents elevation, while $\rho_i$ denotes the latitude and longitude of station $i$. Utilizing topographic data along with specified geographical coordinates, we can determine the elevation of all regions nationwide. By doing so, we can establish a basic graph structure $G$. Upon this foundation, by constructing a Diffusion graph and an Advection graph, and subsequently building GNN Blocks for each, we can approximate the Laplacian operator required by the DAE.

## A.5  DATASETS DESCRIPTION

**Beijing** dataset comprises data from 35 air quality monitoring stations (measuring $PM_{2.5}$, $PM_{10}$, $O_3$, $NO_2$, $SO_2$, and CO), and meteorological reanalysis data (temperature, pressure, humidity, wind speed, and wind direction), organized in a grid format. The dataset contains hourly observations from January 1, 2017, to March 31, 2018. Meteorological data for each station is obtained from the nearest grid point. Missing air quality data is initially filled using information from the nearest station, and gaps of less than 5 hours are interpolated along the time dimension. Gaps exceeding 5 hours are removed to preserve data integrity. The dataset is divided chronologically in a 7:1:2 ratio for training, validation, and testing.

**KnowAir** dataset includes $PM_{2.5}$ data and 17 meteorological attributes from 184 cities across China, with observations recorded at three-hour intervals from January 1, 2015, to December 31, 2018. Unlike the Beijing dataset, the KnowAir dataset is divided chronologically in a 2:1:1 ratio due to its ample four-year data span (Wang et al., 2020).

## A.6  EVALUATION METRICS

Let $x = (x_1, \ldots, x_m)$ represents the ground truth, and $\hat{x} = (\hat{x}_1, \ldots, \hat{x}_m)$ represents the predicted pollutant concentrations. The evaluation metrics we used in this paper are defined as follows:

**Mean Absolute Error (MAE)**

$$\text{MAE}(x, \hat{x}) = \frac{1}{m} \sum_{i=1}^{m} |x_i - \hat{x}_i|$$

**Root Mean Square Error (RMSE)**

$$\text{RMSE}(x, \hat{x}) = \sqrt{\frac{1}{m} \sum_{i=1}^{n} (x_i - \hat{x}_i)^2}$$

**Symmetric Mean Absolute Percentage Error (SMAPE)**

$$\text{SMAPE}(x, \hat{x}) = \frac{1}{m} \sum_{i=1}^{m} \frac{|x_i - \hat{x}_i|}{\frac{|x_i| + |\hat{x}_i|}{2}}$$

## A.7  IMPLEMENTS DETAILS

All experiments are conducted using PyTorch 2.3.0 and executed on an NVIDIA GeForce RTX 3090 GPU, utilizing the Adam optimizer. The batch size is set to 32, and the initial learning rate is 0.005, which decays at specific intervals with a decay rate of 0.1. A GRU-based RNN encoder is employed for the Coefficient Estimator and the encoders of both the Physics Dynamics and Data-Driven Dynamics. For the ODE solver, we adopt the *dopri5* numerical integration method in combination with the adjoint method (Chen et al., 2018). For Dynamics Fusion, $\lambda_1$ and $\lambda_2$ are set to 1 and 0.8, respectively, to differentiate distinct dynamics, and the number of GNN layers is set to 3. The solver's relative tolerance (rtol) and absolute tolerance (atol) are set to 1e-3.

## A.8 NUMERICAL STABILITY

Given that Air-DualODE utilizes an ODE solver in its dual branches and incorporates complex structures like GNN Fusion, it is necessary to verify the model's numerical stability. The training loss curves on the Beijing and KnowAir datasets (Fig. 8) demonstrate that Air-DualODE achieves convergence. Besides, our model incorporates normalization techniques, such as Layer Normalization, which contributes to numerical stability.

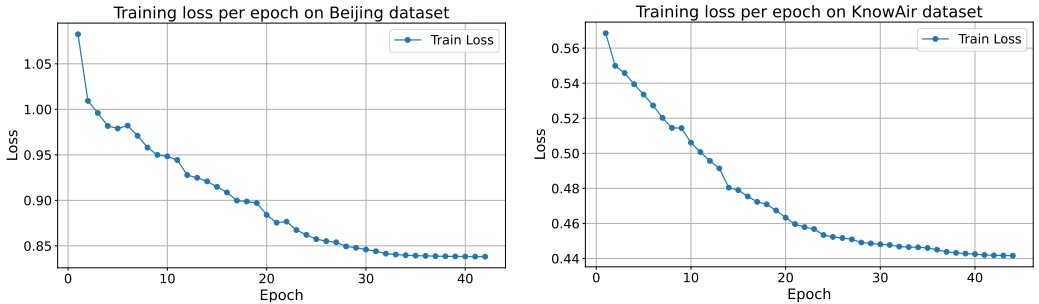

Figure 8: The training curves on the two datasets experimentally demonstrate that Air-DualODE achieves convergence.

## A.9 THE CASE STUDY OF KNOWAIR

Similar to Fig.6, we visualized a sample from the KnowAir dataset. Since KnowAir is a national-level dataset, we selected time points with a two-day interval. Notably, within the purple circle, pollutant concentrations are influenced by the southwest wind, resulting in a gradual decrease in pollutant levels across mainland China. The case studies in Fig.6 and Fig.9 demonstrate that, regardless of the scale of the pollutant prediction scenario, Air-DualODE effectively captures the behavior of the Advection equation, thanks to its dual-branch framework.

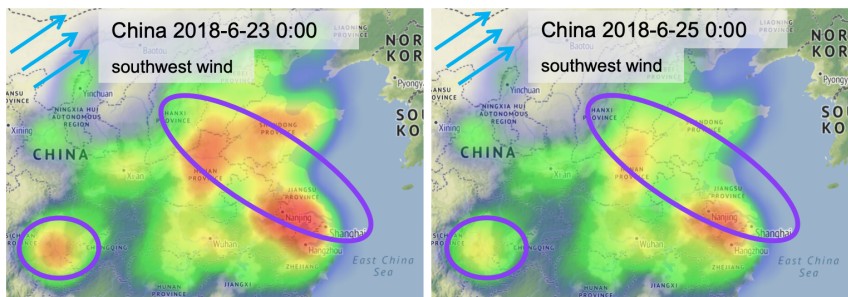

Figure 9: Visualization of predicted $PM_{2.5}$ concentrations under advection. The heatmap shows $PM_{2.5}$ concentration, while arrows represent wind direction.

## A.10 MORE EXPERIMENT RESULTS

To highlight the superiority of Air-DualODE, we provide more comprehensive comparisons by evaluating prediction performance for one-day, two-day, and three-day intervals. Table 4 is more complete than Table 1 in Section 4.2, providing additional short-term prediction results.

At the period-of-time level, from Table 4, it is evident that Air-DualODE outperforms all competitive baselines across all three time intervals within three days. To further evaluate its performance over step error, we provide the MAE results at 12 individual time steps. From Table 5, Air-DualODE

Table 4: More experiment results on Beijing and KnowAir datasets. The bold and underlined font show the best and the second best result respectively.

| Dataset | Model | 1-24h | | | 24-48h | | | 48-72h | | |
|---------|-------|-----|------|-------|-----|------|-------|-----|------|-------|
| | | MAE | RMSE | SMAPE | MAE | RMSE | SMAPE | MAE | RMSE | SMAPE |
| Beijing | LatentODE | 37.28 | 54.31 | 0.70 | 46.08 | 67.11 | 0.83 | 46.67 | 68.82 | 0.84 |
| | PM25GNN | 38.58 | 56.72 | 0.73 | 47.29 | 70.43 | 0.86 | 49.82 | 72.14 | 0.87 |
| | Airformer | 35.88 | 55.01 | 0.69 | 45.62 | 65.61 | 0.81 | 46.73 | 68.69 | 0.83 |
| | AirPhyNet | 35.69 | 58.35 | 0.68 | 45.06 | 66.83 | 0.81 | 47.42 | 68.52 | 0.83 |
| | Air-DualODE | **32.70** | **50.61** | **0.62** | **43.11** | **65.35** | **0.78** | **45.15** | **67.59** | **0.81** |
| KnowAir | LatentODE | 16.82 | 26.93 | 0.39 | 20.56 | 31.35 | 0.47 | 22.59 | 33.86 | 0.52 |
| | PM25GNN | 15.99 | 25.53 | 0.37 | 20.39 | 31.33 | 0.45 | 21.59 | 32.99 | 0.48 |
| | Airformer | 15.63 | 25.51 | 0.36 | 20.37 | 31.52 | 0.45 | 21.49 | 33.02 | 0.47 |
| | AirPhyNet | 17.74 | 27.75 | 0.40 | 21.72 | 32.41 | 0.48 | 24.47 | 34.75 | 0.53 |
| | Air-DualODE | **15.43** | **24.76** | **0.35** | **19.60** | **30.49** | **0.44** | **20.90** | **32.32** | **0.47** |

Table 5: Three-Day MAE Comparison at Each Step on Beijing and KnowAir Datasets. The bold and underlined font show the best and the second best result respectively.

| Dataset | Model | 1st | 3rd | 5th | 7th | 9th | 11th | 13th | 15th | 17th | 19th | 21st | 23rd |
|---------|-------|-----|-----|-----|-----|-----|------|------|------|------|------|------|------|
| Beijing | LatentODE | 28.06 | 34.55 | 39.38 | 42.58 | 44.91 | 45.96 | 46.37 | 46.57 | 46.68 | 46.61 | 46.58 | 46.73 |
| | PM25GNN | 27.45 | 36.11 | 41.59 | 43.09 | 45.34 | 47.33 | 47.85 | 48.37 | 49.62 | 49.66 | 49.45 | 49.33 |
| | Airformer | 23.88 | 31.94 | 38.17 | 41.76 | 44.18 | 45.37 | 45.72 | 45.83 | 46.32 | 46.63 | 46.46 | 46.79 |
| | AirPhyNet | 23.17 | 31.66 | 37.76 | 40.88 | 43.95 | 44.27 | 45.61 | 45.67 | 46.13 | 46.65 | 46.82 | 47.96 |
| | Air-DualODE | **18.57** | **29.68** | **36.00** | **39.80** | **42.00** | **42.85** | **43.21** | **43.76** | **44.39** | **44.80** | **45.23** | **45.71** |
| KnowAir | LatentODE | 14.26 | 16.11 | 17.37 | 18.23 | 19.26 | 20.23 | 20.83 | 21.24 | 21.82 | 22.39 | 22.74 | 22.98 |
| | PM25GNN | 10.45 | 15.60 | 17.35 | 17.98 | 19.03 | 20.30 | 20.82 | 20.85 | 21.08 | 21.66 | 21.82 | 21.65 |
| | Airformer | 10.28 | 14.66 | 16.74 | 18.01 | 19.15 | 19.98 | 20.54 | 20.81 | 21.12 | 21.34 | 21.69 | 21.79 |
| | AirPhyNet | 14.41 | 17.18 | 18.56 | 19.21 | 20.27 | 21.38 | 22.06 | 22.44 | 23.17 | 24.05 | 24.71 | 25.23 |
| | Air-DualODE | **10.26** | **14.65** | **16.59** | **17.66** | **18.54** | **19.33** | **19.82** | **20.17** | **20.47** | **20.79** | **21.02** | **21.12** |

also surpasses all baselines in the step-by-step comparison. These results demonstrate its strong short-term and long-term prediction capabilities.

## A.11 HYPERPARAMETER SENSITIVE ANALYSIS

To examine the robustness of Air-DualODE to different hyperparameters, we selected the number of layers $n$ in the GNN Fusion module (Section 3.4) and the coefficient $\gamma$ in the loss function (Eq.**??**) for sensitivity analysis. As shown in Fig.10 below, both hyperparameters exhibit low sensitivity on the two datasets, achieving consistently good performance.

## A.12 ODE SOLVER SENSITIVE ANALYSIS

To examine the robustness and runtime differences of Air-DualODE across different solvers, we trained the model using three different solvers (Euler, RK4, and Dopri5) on two datasets. The inference time refers to the runtime on the entire test dataset. The results are as follows:

Table 6: Sensitivity and Runtime study of different ODE solver on two datasets.

| Solver | MAE | RMSE | MAPE | Inference time(s) |
|--------|-----|------|------|-------------------|
| Beijing-euler | 41.23 | 63.09 | 0.74 | 0.9 |
| Beijing-rk4 | 40.80 | 62.90 | 0.74 | 1.5 |
| Beijing-dopri5 | **40.32** | **62.04** | **0.74** | 1.98 |
| KnowAir-euler | 18.92 | 30.77 | 0.42 | 11.4 |
| KnowAir-rk4 | 18.94 | 30.42 | 0.42 | 18.9 |
| KnowAir-dopri5 | **18.64** | **29.37** | **0.42** | 22.5 |

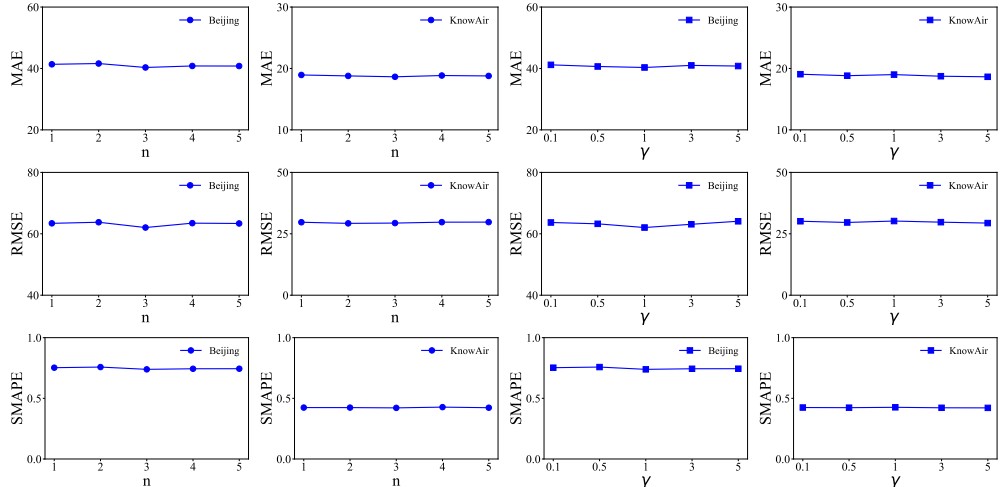

Figure 10: Sensitivity of MAE, RMSE and SMAPE to $n$ and $\gamma$ on Beijing and KnowAir.

As shown in Table 6, Air-DualODE demonstrates a certain level of robustness to different solver types. Dopri5, as an adaptive high-order ODE solver, achieves the best performance on both datasets. Although it incurs a slight increase in runtime, it provides a corresponding improvement in accuracy.

### A.13 VISUALIZATION OF ADVECTION GRAPH

As mentioned in Section 3.2.1, the construction of the Advection Graph is defined in Eq.4. To better understand the dynamic nature of $G_{\text{adv}}$, we visualized the wind speed and direction of several stations in Beijing at different time, along with their corresponding $G_{\text{adv}}$, as shown in Fig.11.

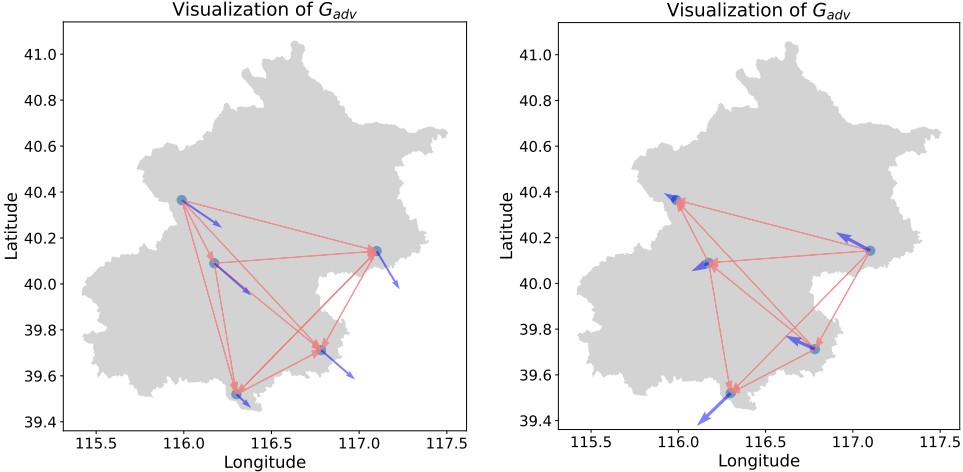

Figure 11: Visualization of $G_{\text{adv}}$ in Beijing. The blue arrows indicate wind direction, and their lengths represent wind speed at each station. The orange arrows illustrate the direct edges of $G_{\text{adv}}$.

In Fig.11, the changes in the Advection Graph depend on variations in wind speed and direction. Specifically, wind direction influences the connectivity between nodes, while wind speed determines the weight of each edge.

## A.14 The details of $\mathbf{F^D}$

$\mathbf{F^D}$ represents the ODE function of the data-driven branch, incorporating a Spatial-MSA structure as shown in Fig.2 ($\mathbf{F^D}$: Masked Attention-Based ODE Fusion). The following section provides details about the formula for $\mathbf{F^D}$.

$$Q, K, V = \text{Projection}(Z^D),$$
$$Z^D = Z^D + \text{Spatial-MSA}(Q, K, V),$$
$$\frac{dZ^D}{dt} = \text{LN}(Z^D + \text{MLP}(\text{LN}(Z^D))) = \mathbf{F^D}.$$

Among these, $Q$, $K$, and $V$ are obtained through a linear projection of $Z^D$. The spatiotemporal dependencies are then captured using the carefully designed Spatial-MSA and a residual connection in the form of a data-driven derivative. Layer normalization is applied to ensure numerical stability.

We consider air pollutant propagation as a spatiotemporal dynamic system. Neural ODE, serving as a bridge between dynamic systems and neural networks, is better suited for modeling the spatiotemporal dynamics of pollutants compared to traditional sequence models like RNN and Transformer.

## A.15 Visualization of sudden changes' results

To validate whether Air-DualODE can predict upward or downward trends during sudden changes (as described in Section 4.1), we visualized sudden changes at specific stations, as shown in Fig.12. The visualization results demonstrate that Air-DualODE can predict the overall upward and downward trends, providing important support for downstream decision-making by policymakers. However, in some cases, such as those illustrated in Fig.12a and Fig.12b, the peak pollutant concentrations were not well predicted. Although Air-DualODE outperforms other methods in the quantitative analysis of sudden changes prediction, as shown in Table 1, predicting sudden changes remains a significant challenge.

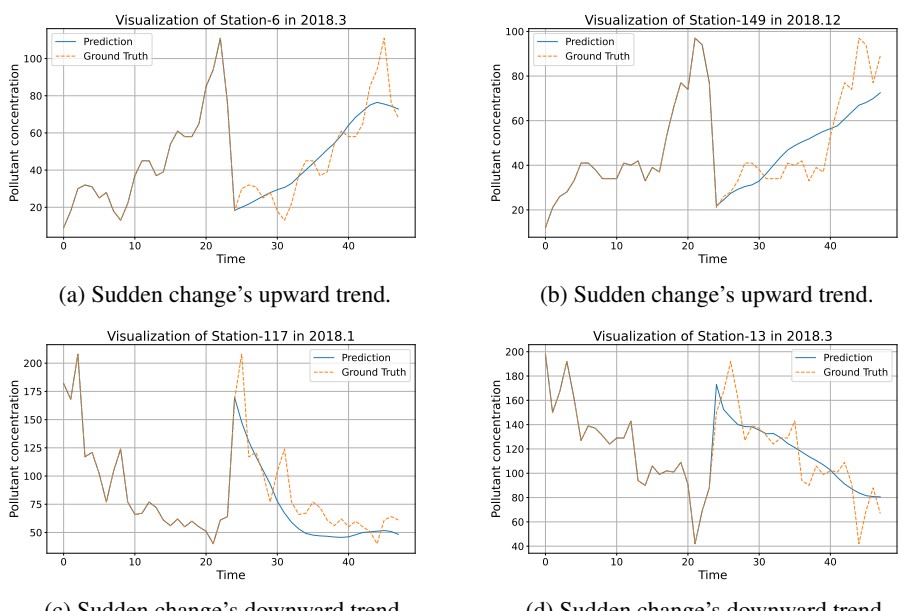

Figure 12: Sudden change visualization on KnowAir dataset.

## A.16 ABLATION STUDY ON GNN FUSION

To experimentally demonstrate the effectiveness of GNN Fusion, we replaced GNN Fusion with other variants. Among these, MLP Fusion maps from $\mathbb{R}^{N \times 2D} \to \mathbb{R}^{N \times 1}$, while Equivalent MLP Fusion uses an equivalent number of MLP layers as GNN Fusion. Additionally, we tested Pooling Fusion, which applies mean or sum pooling to each station's representations. The results are presented in Table 7.

Table 7: Effect of GNN Fusion in Dynamics Fusion.

| Methods | Beijing | | | KnowAir | | |
|---|---|---|---|---|---|---|
| | MAE | RMSE | SMAPE | MAE | RMSE | SMAPE |
| MLP Fusion | 41.49 | 63.14 | 0.76 | 21.50 | 32.92 | 0.56 |
| Equivalent MLP Fusion | 41.45 | 63.23 | 0.75 | 21.42 | 32.83 | 0.56 |
| Mean-Pooling Fusion | 43.23 | 68.88 | 0.80 | 20.03 | 31.22 | 0.45 |
| Sum-Pooling Fusion | 43.12 | 68.30 | 0.80 | 19.97 | 31.10 | 0.44 |
| Air-DualODE | **40.32** | **62.04** | **0.74** | **18.64** | **29.37** | **0.42** |

According to Table 7, we observe that both MLP-based Fusion and Pooling Fusion exhibit degradation on the Beijing dataset because the Beijing dataset is city-level, where stations are relatively close to each other. Since Pooling Fusion does not consider representations from other stations, its performance degrades more significantly than that of MLP-based Fusion. Conversely, on the KnowAir dataset, we observe that MLP-based fusion methods exhibit significant degradation. This is because, during fusion, these methods incorporate representations from distant nodes, which act as noise for predicting the target station. Such noisy representations fail to provide useful information and instead interfere with prediction accuracy. Additionally, since Pooling fusion methods do not consider representations from other stations, its results outperform the two MLP-based fusion methods. These two extreme fusions highlight the importance of incorporating representations from neighboring stations during fusion. Based on this intuition, we designed GNN Fusion, which leverages the geospatial graph $G$ to fuse representations from nearby stations. The results in Table 7 demonstrate this design choice is effective.

## A.17 DISCUSSIONS

**Sudden change Evaluation.** We use the current definition of sudden changes and the corresponding metrics to ensure consistency with existing studies (Liang et al., 2023; Hettige et al., 2024). However, they may not be the most appropriate ones. In future work, we plan to explore alternative evaluation metrics, such as mean directional accuracy (MDA) (Van den Burg & Williams, 2020), and adopt new sudden change definitions based on change point detection and anomaly detection algorithms (Witzke et al., 2023; Shentu et al., 2025; Wu et al., 2025).

**Numerical instability when removing layer normalization.** In our experiments, we observe that Air-DualODE's numerical stability relies on Layer Normalization when using the **Dopri5 ODE solver**, an adaptive solver in Neural ODE. Without it, gradient explosion and NaN occur after a few training epochs, preventing the model from converging stably. This dependency on Layer Normalization is closely related to solving the dual dynamics with the adaptive ODE solver, as we do not observe this phenomenon when using either a single dynamics branch alone or simpler, fixed-step ODE solvers (e.g., Euler and RK4). The dual dynamics, with their inconsistent numerical ranges, lead to instability during adaptive forward and backward propagation. However, Layer Normalization ensures consistency of the numerical ranges in the dual dynamics' adaptive solving during each iteration, thereby ensuring numerical stability.

**Comparison with AirPhyNet.**

- Air-DualODE models open air systems, making it more realistic compared to the closed system assumption of AirPhyNet.

Table 8: The difference between Air-DualODE and AirPhyNet.

| Difference | AirPhyNet | Air-DualODE |
|---|---|---|
| Air system modeling | Closed system | Open system |
| Physics-guided approaches | Explicit physical equation on latent space | Explicit physical equation on explicit space and GNN fusion |
| Matching physics with latent representations | No mechanism to ensure | Decay-TCL |
| Spatial module | GNN-based differential equation | NODE with Spatial-MSA and BA-DAE |
| Temporal module | GNN-based differential equation | NODE and BA-DAE |
| Computational efficiency | Slow | Fast |

- Air-DualODE applies physical equations directly to explicit variables, which is more reasonable than applying them to latent variables that no longer hold actual physical meanings.

- Air-DualODE employs Decay-TCL for alignment to address the mismatch between explicit physical equations and data-driven latent representations, whereas AirPhyNet overlooks this issue.

- AirPhyNet relies only on GNN-based differential equations to model spatiotemporal dependencies, whereas Air-DualODE incorporates multiple components, including BA-DAE for modeling in the explicit space and NODE with Spatial-MSA for capturing additional dependencies in the latent space.

- Air-DualODE outperforms AirPhyNet in terms of computational efficiency and scalability to a larger number of nodes.

### A.18 LIMITATIONS

**Generalization to other domains.** In this paper, Air-DualODE is specifically designed for air pollutant prediction. To apply it to spatiotemporal prediction tasks in other domains (Guo et al., 2014; Liu et al., 2018) (e.g., traffic forecasting and water quality prediction), the physical branch would require adaptive modifications.

**Handling sudden changes.** As mentioned in Appendix A.15, predicting sudden changes remains a challenge in air quality prediction. This is an issue that demands significant attention because sudden changes in pollutant concentrations are critical to societal activities and production. In fact, many spatiotemporal dependencies underlying sudden changes cannot be effectively captured by deterministic methods or physical equations, as such changes are often influenced by uncertainties in weather and human interventions. In the future, we plan to explore and address this challenge further by incorporating probabilistic approaches like variational inference into the data-driven branch. We leave this as an important and promising future research direction.

