# OpenReview forum: "Air Quality Prediction with Physics-Guided Dual Neural ODEs in Open Systems"
_ICLR.cc/2025/Conference — ICLR 2025 Poster_

### Official Review · Reviewer_iuiU · 2024-10-25

**Soundness:** 3
**Presentation:** 3
**Contribution:** 2
**Rating:** 6
**Confidence:** 4

**Summary:**

This paper presents Air-DualODE, a hybrid model that integrates physics-based and data-driven techniques. The proposed model addresses the limitations of traditional physics-based and data-driven models, particularly the assumptions about closed systems and the mismatch between explicit physical equations and learned representations. The model involves two main components, where one component applies open-system physical equations to model spatiotemporal dependencies and learn physical dynamics, while the second component identifies the unmodeled dependencies using a fully data-driven approach. The model achieves superior performance  on two real-world datasets at city scale and national scale, compared to the other baselines in this domain.

**Strengths:**

1. The paper addresses a significant and timely problem.
2. The write-up style and the narrative of the paper is clear, structured, and professional.
3. The reported results are very strong compared to the existing baselines.

**Weaknesses:**

1. The paper does not adequately address the scalability of the proposed model. Given that the approach involves multiple components like ODE solvers and GNNs, the computational cost could become prohibitive when applied to larger datasets with thousands of locations. (ex: Dataset used in AirFormer (Liang et al., 2023)).

2. The overall approach and the methodology of this work closely resembles prior work like AirPhyNet (Hettige et al., 2024), which also integrates physics-based diffusion-advection processes with neural networks for air quality prediction, raising concerns about novelty and significant contributions.

3. Eventhough the work emphasizes about the interpretability of the model, there is limited explanation about how the model’s output could be interpreted in terms of real-world physical phenomena.

4. All the baselines used in the experiments are either fully data driven or hybrid models. Advanced physics based models are not included as baselines and the performance is not compared with the proposed model. Can you evaluate your model’s performance against specific physics-based models, such as Community Multiscale Air Quality (CMAQ) , Weather Reaserch Forecasting (WRF), AERMOD to provide a more comprehensive understanding of how your hybrid model compares in terms of accuracy and computational efficiency?

**Questions:**

1. The introduction and role of the $\beta$ term in Equation 6 is not fully clear. Could you provide more details including mathematical justifcation on how this term is derived and its exact significance within the  framework?

2. Could you provide more theoretical justification and intuition behind the Decay-TCL mechanism? Specifically, how do the hyperparameters influence the alignment and fusion process between the physics-based and data-driven dynamics?

3. Could you elaborate  on how the GNN fusion mechanism effectively balances the two components?

4. How scalable is Air-DualODE to larger datasets with thousands of nodes? Could you provide a detailed complexity analysis  of your model as the number of locations (nodes) increases ?

5. Could you provide more clarification on how the predictions can be interpreted and provide specific examples on how these outputs help policy makers or environmental scientists in their decision making process?

6. The proposed model and AirPhyNet (Hettige et al., 2024) employ a similar hybrid architecture of physics based and data driven models , combining GNNs and differential equation solvers. Additionally, both approaches emphasize interpretability through case studies that link predictions to real-world physical phenomena, such as pollutant dispersion influenced by environmental factors like wind speed and direction. Could you elaborate on how your model differs methodologically and in its interpretive capabilities and what specific aspects of your work provide novelty beyond what is addressed in AirPhyNet?

---

> ### Author Response · Authors · 2024-11-21
> **Response to Reviewer iuiU**
>
> Thank you for the insightful review. We appreciate the acknowledgment of our novel architecture and strong results. In response to your feedback, the noted concerns are addressed below.
>
>
> ### **W1, Q4**: the scalability of Air-DualODE and concerns about the computational cost
>
> That is a good question! This is precisely why we conducted experiments on the KnowAir dataset, which represents a national-level air quality prediction scenario. From our perspective, the computational cost of Air-DualODE should remain manageable even when applied to larger datasets. Unfortunately, there does not exist publicly available datasets with larger sizes. While Airformer utilizes a nationwide dataset with 1,085 stations, this dataset is not publicly accessible online. To address this, we generate a synthetic dataset with 1085 nodes to evaluate the method’s scalability on large-scale datasets.
>
> We selected AirPhyNet as a competitor because both AirPhyNet and Air-DualODE belong to the family of physics-guided machine learning approaches. Here is their training and inference time costs (one epoch):
>
> * Training Time
>
> |                 | AirPhyNet               | Air-DualODE |
> | --------------- | ----------------------- | ----------- |
> | Beijing(35)     | 90s                     | 81s         |
> | KnowAir(184)    | 3870s                   | 378s        |
> | Synthetic(1085) | larger than three hours | 1235s       |
>
> * Inference Time
>
> |                 | AirPhyNet              | Air-DualODE |
> | --------------- | ---------------------- | ----------- |
> | Beijing(35)     | 2.75s                  | 1.98s       |
> | KnowAir(184)    | 540s                   | 22.5s       |
> | Synthetic(1085) | larger than 30 minutes | 54s         |
>
> For fair comparison, these experiments are conducted on **a single** NVIDIA A800 Tensor Core GPU.
>
> AirPhyNet involves many MLP structures (e.g., Gated Fusion) tied to the number of stations during the solving process. The gradients of these parameters are tracked and recorded in the GPU during training, leading to an increase in the computational cost of the ODE solver as the number of stations grows. In contrast, Air-DualODE incorporates several optimizations, such as Spatial-MSA, which effectively reduce the overall computational complexity. Therefore, despite utilizing two ODE solvers, the overall computational cost of Air-DualODE remains manageable.
>
> ### **W2, Q6**: the difference between AirPhyNet and Air-DualODE
>
> The main differences are as follows:
>
> | Difference                                       | AirPhyNet                                  | Air-DualODE                                                 |
> | ------------------------------------------------ | ------------------------------------------ | ----------------------------------------------------------- |
> | **Computational efficiency**                     | Slow                                       | Fast                                                        |
> | **Air system modeling**                          | Closed system                              | Open system                                                 |
> | **Physics-guided approaches**                    | Explicit physical equation on latent space | Explicit physical equation on explicit space and GNN fusion |
> | **Matching physics with latent representations** | No mechanism to ensure                     | Decay-TCL                                                   |
> | **Spatial module**                               | GNN-based differential equation            | NODE with Spatial-MSA and BA-DAE                            |
> | **Temporal module**                              | GNN-based differential equation            | NODE and BA-DAE                                             |
>
> 1. As discussed in W1, Q4, Air-DualODE outperforms AirPhyNet in terms of computational efficiency and scalability to a larger number of nodes.
> 2. Air-DualODE models open air systems, making it more realistic compared to the closed system assumption of AirPhyNet.
> 3. Air-DualODE applies physical equations directly to explicit variables, which is more reasonable than applying them to latent variables that no longer hold actual physical meanings.
> 4. To address the mismatch between explicit physical equations and data-driven latent representations, Air-DualODE employs Decay-TCL for alignment, whereas AirPhyNet overlooks this issue.
> 5. AirPhyNet relies only on GNN-based differential equations to model spatiotemporal dependencies, whereas Air-DualODE incorporates multiple components, including BA-DAE for modeling in the explicit space and NODE with Spatial-MSA for capturing additional dependencies in the latent space.

---

> ### Author Response · Authors · 2024-11-21
> **Response to Reviewer iuiU**
>
> ### **W3, Q5**: the prediction's interpretability for policy maker
>
> Our model can provide stakeholders with explanations to some extent. In Section 4.4, the second case study presents a visualization of $\boldsymbol{\beta}$ under two scenarios in Beijing. In our view, $\boldsymbol{\beta}$ can serve as an important reference indicator for stakeholders: $\beta_i < 0$ indicates a sink (e.g., out-of-boundary diffusion/advection or forest absorption), while $\beta_i > 0$ indicates a source (e.g., industrial activity or vehicular emissions). This provides stakeholders with insights into how the model makes predictions to some degree.
>
>
>
> ### **W4**: Air-DualODE Compared with CMAQ, WRF and AERMOD
>
> We appreciate your observation regarding the absence of a direct comparison with physics-based models such as CMAQ (Community Multiscale Air Quality) and WRF (Weather Research and Forecasting). These models require numerous input variables, including land use data, boundary conditions, meteorological parameters (e.g., boundary layer height) which are not available for our dataset. Thus, we are unable to run such physics based models for comparisons.
>
>
> ### **Q1**: mathematical intuition on $\boldsymbol{\beta} X$ and significance within the framework
>
> **mathematical intuition on $\boldsymbol{\beta} X$**
>
> This term is motivated from the Infectious differential equation in SIR model.
> $$
> \begin{align*}
> \frac{dI}{dt} &= \eta \frac{SI}{N} - \gamma I  \\\\
> \frac{dR}{dt} &= \gamma I
> \end{align*}
> $$
>
> * **Susceptible (S), Infectious (I), Recovered (R)**
> * $N = S + I + R$ is the total population.
> * $\eta$ is the transmission rate and $\gamma$ is the recovery rate.
>
> In these differential equations, the growth rate of recovery is modeled as a proportion to infected population. However, in reality, their relationship exhibits complex nonlinear dependencies. Motivated by this design and the principle of Occam’s razor, we also use $\boldsymbol{\beta} X$ to model sources or sinks dynamics, which are crucial in an open air system.
>
> **significance within the framework**
>
> Combined with Eq.5 in Section 3.2.1, it can be observed that BA-DAE no longer satisfies mass conservation, which is more realistic for an open air system. As mentioned in Section 3.2.2, the dissipation of pollutants cannot exceed the current pollutant concentration; therefore, $\boldsymbol{\beta}$ has a minimum value of -1, i.e., all pollutants are dissipated. However, the generation of pollutants may far exceed the current concentration. Thus, $\boldsymbol{\beta}$ can theoretically have a maximum value of $+\infty$. Besides, we perform an ablation study to evaluate the significance of $\boldsymbol{\beta} X$ in open air system. In Fig.5 DAE-c without $\boldsymbol{\beta} X$ is for closed system modeling, while BA-DAE-o with $\boldsymbol{\beta} X$ consider the open air system. The results highlights the significance of $\boldsymbol{\beta} X$ in our framework.

---

> ### Author Response · Authors · 2024-11-21
> **Response to Reviewer iuiU**
>
> ### **Q2**: theory and intuition behind Decay-TCL mechanism and the hyperparameters' influence to the alignment and fusion process
>
> **theory and intuition behind Decay-TCL mechanism**
>
> Decay-TCL is derived from temporal contrastive learning techniques. In Section 3.4, $\lambda_1$ controls the soft weight distribution for intra-dynamics latent representations, while $\lambda_2$ controls the soft weight distribution for inter-dynamics latent representations. These hyperparameters determine the sharpness of $w(t, s)$. The similarity of intra-dynamics latent representations should decay along the time axis, while the decay of inter-dynamics latent representations should account for the differences between two types of dynamics, which satisfies the rule1 and rule2 (rf. line323-324). It is reasonable to enforce $\lambda_1 > \lambda_2$, as intra-dynamics latent representations should exhibit greater similarity than inter-dynamics, satisfying Rule 3 (ref. line 325). By adhering to these rules, the soft InfoNCE $\ell(t)$ in Eq.11 adjusts the similarity of the two dynamics’ latent representations effectively.
>
> **the hyperparameters' influence to the alignment and fusion process**
>
> If both $\lambda$ are very large, the distribution of $w(t, s)$ becomes sharper, significantly reducing the number of soft positive samples. This forces the model to focus more on learning the differences between the dynamics’ latent representations. However, dynamics inherently exhibit continuity, and emphasizing large differences can disrupt this continuity. This negatively impacts ODE solver-based learning and breaks the alignment between latent representations.
>
> Conversely, if both $\lambda$ are very small, the distribution of $w(t, s)$ becomes smoother, reducing the number of negative samples. This causes the model to focus more on the commonalities between the dynamics’ latent representations. However, since physics-based dynamics already capture spatiotemporal dependencies, this approach may prevent data-driven dynamics from learning additional spatiotemporal dependencies in the latent space, rendering the subsequent fusion process ineffective.
>
> Therefore, considering these impacts and the constraint $\lambda_1 > \lambda_2$, we conducted a grid search (restricted to the upper or lower triangular region) rather than relying on experience, ultimately obtaining $\lambda_1 = 1$ and $\lambda_2 = 0.8$.
>
>
>
> ### **Q3**: how the GNN fusion mechanism effectively balances the two components?
>
> First, before applying the GNN fusion, the Decay-TCL are used to map the representations into a shared latent space as much as possible, aligning physics and data-driven dynamics latent representations. Second, the GNN layers adjust the influence of individual nodes based on their spatial connectivity, allowing the model to balance information across geographically connected stations.

---

> > ### Comment · Reviewer_iuiU · 2024-11-25
> >
> > Thank you for your detailed and thorough elaborations on my concerns. I have a few follow-up questions:
> >
> > **Scalability:**
> > 1. While you emphasize optimizations like Spatial-MSA, the explanation does not include a formal complexity analysis of how the computational cost scales with the number of nodes. Could you provide a formal complexity analysis (e.g., \( O(n) \) or \( O(n^2) \)) for the major components of Air-DualODE, particularly the ODE solvers and Spatial-MSA? How do these scale with increasing numbers of nodes?
> >
> > **Interpretability:**
> > 1. Could you elaborate on how $\beta$ is computed and provide a clearer explanation of its physical meaning?
> > 2. Beyond identifying sources and sinks, how can the model’s outputs inform specific policies or interventions? How are these outputs communicated to stakeholders in practice?
> >
> > **GNN Fusion and Decay-TCL:**
> > 1. Have you conducted experiments with alternative fusion approaches, such as weighted averaging or attention mechanisms, in addition to GNN fusion?
> > 2. Can Decay-TCL and the GNN fusion mechanism be generalized to other spatiotemporal tasks, such as traffic forecasting or climate modeling? If so, what modifications, if any, would be required to adapt them to different domains?

---

> ### Author Response · Authors · 2024-11-25
> **Response to Reviewer iuiU**
>
> ## Scalability
>
> ### **Q1**: complexity analysis
> We provide a formal complexity analysis for the major components of Air-DualODE. In physical branch, the complexity is $O(I_P \cdot |E|)$. In the data-driven branch with Spatial-MSA, the complexity is $O(I_D \cdot M^2 \cdot C)$. Here, $I_P$ and $I_D$ refer to the iteration time for the physics and data-driven dynamics, respectively. $|E|$ is the number of edge in the $G$. $C$ represents the dimension of latent variables. $M$ represents number of neighboring stations around the current station. When scaling with an increasing number of nodes, the complexity growth primarily depends on the number of neighboring nodes for each node rather than the total number of nodes.
>
> ## Interpretability
>
> ### **Q2**: computation of $\boldsymbol{\beta}$ and its physical meaning
>
> The $\boldsymbol{\beta}$ is estimated according to the historical data: $T \times N \times D$, where $T$ means the historical timestamps, $N$ means the number of stations and $D$ means the number of input variables. The mapping of $\boldsymbol{\beta}$ can be defined as $\mathbb{R}^{T \times N \times D} \to \mathbb{R}^{N \times 1}$ using the RNN-based coefficient estimator, which assigns a specific $\beta_i \in \mathbb{R}$ to each station $i$.
>
> Regarding the physical meaning: $\beta_i > 0$  indicates that station $i$ is likely influenced by a source nearby like industrial activity and vehicular emissions , while $\beta_i < 0$ suggests that station $i$ is likely influenced by a sink like forest absorption.
>
> ### **Q3**: Beyond identifying sources and sinks, how can the model’s outputs inform specific policies or interventions? How are these outputs communicated to stakeholders in practice?
>
> Because each time window estimates $\boldsymbol{\beta}$, if the estimation consistently results in $\beta_i < 0$ over a continuous period, it indicates the presence of a sink near station $i$ that absorbs pollutants during that time. This provides stakeholders with insights or suggestions on how to create sinks to absorb pollutants. Conversely, if the estimation consistently results in $\beta_i > 0$ over a continuous period, it signifies the presence of a source near station $i$ generating pollutants. In such cases, policymakers may need to implement regulations to reduce $\beta_i$, mitigating pollutant generation.
>
>
> ## GNN Fusion and Decay-TCL
>
> ### **Q4**: alternative fusion approaches
>
> Yes, we conducted experiments with alternative fusion approaches, see the Table below. The results show that other fusion approaches are not as effective as our GNN Fusion.
>
> | Model                                | Beijing |       |       | KnowAir |        |       |
> | ------------------------------------ | ------- | ----- | ----- | ------- | ------ | ----- |
> |                                      | MAE     | RMSE  | SMAPE | MAE     | RMSE   | SMAPE |
> | MLP Fusion (one layer)               | 41.49   | 63.14 | 0.76  | 21.5    | 32.919 | 0.56  |
> | Equivalent MLP Fusion (three layers) | 41.45   | 63.23 | 0.75  | 21.42   | 32.83  | 0.56  |
> | Mean-Pooling Fusion                  | 43.23   | 68.88 | 0.80  | 20.03   | 31.22  | 0.45  |
> | Sum-Pooling Fusion                   | 43.12   | 68.30 | 0.80  | 19.97   | 31.10  | 0.44  |
> | Air-DualODE                          | 40.32   | 62.04 | 0.74  | 18.64   | 29.37  | 0.42  |
>
> **Note:** MLP fusion introduces representations from distant stations, which act as noise for the current station. As a result, performance degradation occurs when GNN fusion is replaced with MLP fusion.
>
> ### **Q5**: generalization to traffic forecasting or climate modeling
>
> To clarify, Decay-TCL and the GNN fusion mechanism are specifically designed for Air-DualODE due to the physical and data-driven dynamics. The BA-DAE in the physical branch may not align with traffic forecasting or climate modeling because their governing equations are different. Therefore, Air-DualODE is not directly applicable to these tasks. However, by revising the Air-DualODE's physical branch with domain specific equations (e.g., Navier-Stokes equations for climate), we believe it has the potential to generalize to those domains.

---

> > ### Comment · Reviewer_iuiU · 2024-11-26
> >
> > Thank you for your clarifications to my questions/comments. I updated my rating to a 6 and hold a positive outlook on the paper.

---

> > > ### Author Response · Authors · 2024-11-26
> > >
> > > Thank you sincerely for your thoughtful feedback and for recognizing our work. Best wishes!

---

### Official Review · Reviewer_joi5 · 2024-11-02

**Soundness:** 3
**Presentation:** 3
**Contribution:** 3
**Rating:** 6
**Confidence:** 3

**Summary:**

The authors propose a model called Air-DualODE that integrates dual branches of Neural Ordinary Differential Equations (ODEs) to enhance air quality predictions. To address the mismatch between explicit physical equations and implicit learned representations, the paper introduces a hybrid model consisting of two branches.

The Physics Branch utilizes the Boundary-Aware Diffusion-Advection Equation (BA-DAE) to capture the spatiotemporal dependencies of pollutants. Meanwhile, the Data-Driven Branch captures dependencies not addressed by the physical equations. These dual representations are skillfully fused using a dynamics fusion module with decaying temporal alignment, which enhances prediction accuracy.

The effectiveness of this method is demonstrated on two widely-used air quality prediction datasets, showing that the model outperforms other existing methods while offering strong interpretability compared to purely data-driven approaches.

**Strengths:**

- Enhanced Physical Modeling: This paper innovatively transforms closed-system physical equations into open-system physical equations based on existing Physics-Informed Neural Networks (PINNs). This modification aligns more closely with real-world conditions and improves interpretability in boundary regions.

- Effective Integration of Knowledge: The DualODE model skillfully integrates physical knowledge with data-driven methods to bridge the gap between physical equations and real-world data. This dual approach ensures that the model captures dependencies that are not addressed by physical equations alone.

- Advanced Fusion Technique: By employing dynamic fusion to combine the representations from the two ODE models, and further refining the output through a Graph Neural Network (GNN), the model effectively captures real-world patterns. This approach outperforms the simplistic method of directly concatenating the two representations, leading to more accurate and realistic predictions.

**Weaknesses:**

- Insufficient Detail on Data-Driven Neural ODE: The paper does not provide a detailed description of the data-driven Neural Ordinary Differential Equations (ODE). Additionally, the necessity of using a Neural ODE model as the data-driven approach is not clearly justified, which may leave readers questioning its selection over other potential methods.

- Limited Results Presentation: The results presented in the paper are limited to predictions for 3 days and scenarios of sudden changes. Predictions for shorter time frames, such as 1-day and 2-day forecasts, are not included. Furthermore, there are discrepancies between the results reported for the reproduction of other works and those in the original publications, which could undermine the credibility of the comparative analysis.

- Language Errors in Figures and Tables: There are minor language errors in the figures and tables that could cause confusion. These errors necessitate careful cross-referencing with the accompanying text to ensure clarity and accurate interpretation of the data.

**Questions:**

- Errors in Figure and Table Text: There are textual and informational errors in the figures and tables. In Figure 2, "Pollutant Contentration" should be corrected to "Pollutant Concentration," and there is an error in the expression for Wind speed. Additionally, the last row in Table 2 should be labeled as "Air-DualODE." without “w/o”.

- Incomplete and Redundant Equations in the Appendix: The equations following lines 690-692 in the appendix are incomplete and appear to be redundant with the content in lines 219-220 of the main text. Could these be revised for clarity and completeness?

- Detailed Derivation of Equation 3 and $F^D$: It would be beneficial to provide a more detailed derivation of Equation 3 in the appendix to enhance understanding. Additionally, the $F^D$ formula is not provided. Could you include a more thorough description to facilitate reader comprehension?

---

> ### Author Response · Authors · 2024-11-21
> **Response to Reviewer joi5**
>
> We would like to sincerely thank Reviewer joi5 for acknowledging our work's novelty and contributions. Your curiosity about Data-Driven Neural ODE and our experiment results are resolved in following contents. Some incomplete contents and typos have been revised in the updated version.
>
>
> ### **W1, Q3**: details about Data-Driven Neural ODE and other potential methods to replace
>
> **Details about Data-Driven Neural ODE**
>
> $F^D$ is a data-driven branch's ODE function with Spatial-MSA Structure. We have provided the details of $F^D$ in the Appendix $\underline{\text{A.14 The details of $\mathbf{F^D}$}}$. The following contents are about $\mathbf{F^D}$ formula.
> $$
> \begin{align*}
>     Q, K, V &= \text{Projection}(Z^D), \\\\
>     Z^D &= Z^D + \text{Spatial-MSA}(Q, K, V), \\\\
>     \frac{d Z^D}{dt} &= \text{LN}(Z^D + \text{MLP}(\text{LN}(Z^D))) = \mathbf{F^D}.
> \end{align*}
> $$
> Among these, $Q$, $K$, and $V$ are obtained through a linear projection of $Z^D$. The spatiotemporal dependencies are then captured using the carefully designed Spatial-MSA and a residual connection in the form of a data-driven derivative. Layer normalization is applied to ensure numerical stability.
>
> **other potential methods to replace**
>
> We consider air pollutant propagation as a spatiotemporal dynamic system. Neural ODE, serving as a bridge between dynamic systems and neural networks, is better suited for modeling the spatiotemporal dynamics of pollutants compared to traditional sequence models like RNN and Transformer.
>
>
> ### **W2**: more results on 1-day and 2-day forecasts and discrepancies between our results and previous works
>
> We have provided the 1-day and 2-day results in the updated version's Appendix $\underline{\text{A10. More experiment results}}$.
>
> **discrepancies between our results and previous works**
>
> At the dataset level, the preprocessing methods used in other studies differ from ours. Due to missing values in the raw data of the Beijing dataset, previous work used linear interpolation to fill in the gaps over time. However, this approach neither incorporates data from other stations at the same time nor adequately addresses the unrealistic assumption of linearity in cases of large consecutive missing intervals. Our preprocessing approach considering these two problems is described in detail in Appendix $\underline{\text{A.5 Dataset Description}}$. At the experimental design level, the settings are also different. For example, in the KnowAir dataset, PM25GNN uses wind speed and direction at the next time step as covariates for all models. However, we believe this design leaks future data, which may deviate from real-world pollutant concentration prediction scenarios. To ensure reproducibility, we have provided an anonymous repository link containing our code (cf. abstract line 027).
>
> ### **W3, Q1**: Language Errors in Figures and Tables
>
> Thank you for pointing out these issues. We have revised in the updated version.
>
>
>
> ### **Q2**: Incomplete and Redundant Equations in the Appendix
>
> We have revised the math part in the Appendix and given more details about diffusion equation and advection equation in the updated version's Appendix $\underline{\text{A.2 The Derivation of Diffusion-Advection Equation}}$.

---

> ### Author Response · Authors · 2024-11-25
> **Looking forward to your feedback**
>
> Dear Reviewer joi5,
>
> Thank you for your valuable and constructive reviews, which has inspired further improvements to our paper. We have made an extensive effort to address your questions and concerns by providing additional results for 1-day and 2-day forecasts, revising our paper and complementing the details of the equations in the Appendix. We hope our response can effectively address your concerns, If you have any further concerns or questions, please do not hesitate to let us know, and we will respond timely.
>
> Best regards,
>
> Authors

---

> > ### Comment · Reviewer_joi5 · 2024-11-26
> >
> > Thank you for your detailed response and for addressing the concerns raised in my initial review. I keep my current Accept score.

---

> > > ### Author Response · Authors · 2024-11-26
> > >
> > > Thank you sincerely for your thoughtful feedback and for recognizing our work. Best wishes!

---

### Official Review · Reviewer_v3np · 2024-11-03

**Soundness:** 4
**Presentation:** 3
**Contribution:** 4
**Rating:** 8
**Confidence:** 4

**Summary:**

To enhance air quality prediction in open systems, this paper proposes a dual Neural ODE architecture named Air-DualODE. It combines a boundary-aware diffusion-advection Equation (BA-DAE) for physical dynamics with a Neural ODE employing masked spatial attention for data-driven dynamics. The framework also incorporates a fusion mechanism that temporally aligns and merges outputs from these dynamics in a shared latent space. Experimental results on both city- and national-level datasets demonstrate state-of-the-art performance.

**Strengths:**

1.This paper addresses a significant problem, i.e. air quality prediction in open systems. In particular, the model thoroughly considers the non-conservation of pollutant concentration within the region of interest, introducing the BA-DAE to effectively model sources and sinks within the area.
2.The author propose an interesting Air-DualODE framework that models known physical equations and unknown spatiotemporal dependencies separately, then aligns and fuses them in the latent space. This approach highlights the guiding role of physical knowledge within the model while allowing it to capture unknown spatiotemporal dependencies that may not be described by physical equations.
3.In experiments, the model achieves superior performance across multiple metrics on different spatial scales datasets. Besides, the provided code enhance this paepr’s reproducibility.
4.The writing and structure of this paper are clear and easy to understand.

**Weaknesses:**

1.The discrete diffusion and advection equations are not thoroughly described in this paper. Highly recommend fully explaining in the appendix.
2.Please elaborate the role of spatial-MSA in data-driven branch presented in Section 3.3.
3.Check some typos in this paper. For example, in Section 4.2 Table 1, "Beijing1718" should be “Beijing”.

**Questions:**

1.Why using the GNN Fusion after temporal alignment? What about other simple structure like MLP?
2.Is there other data should be included for geospatial graph construction?

---

> ### Author Response · Authors · 2024-11-21
> **Response to Reviewer v3np**
>
> Thank you very much for recognizing our idea and contributions. The shortcomings you mentioned have been addressed in the updated version.
>
>
>
> ### **W1**: necessary mathematic description in Appendix
>
> Thank you for your suggestion. We have added discrete diffusion and advection equations and their explanations in the updated version's Appendix $\underline{\text{A.2 The Derivation of Diffusion-Advection Equation}}$ and $\underline{\text{A.3 Proof of closed system's conservation property}}$.
>
>
>
> ### **W2**: the role of spatial-MSA in data-driven branch
>
> The purpose of Spatial-MSA is to exclude distant nodes from the computation of attention scores, preventing interference and noise from affecting the current station. This is because we aim for the data-driven branch to focus solely on capturing the spatiotemporal dependencies of adjacent nodes.
>
>
>
> ### **W3**: some typos in our paper
>
> We have corrected all typos in the updated version.
>
>
>
> ### **Q1**: Why using the GNN Fusion after temporal alignment? What about other simple structure like MLP?
>
> GNN Fusion leverages a distance-based graph structure. While this design may not significantly improve performance on the Beijing dataset (because these stations are close to each other), it proves highly effective for the national-level KnowAir dataset. Intuitively, nodes that are closer together should interact with other stations' dynamics latent representations, whereas distant nodes do not require such interactions. Therefore, GNN Fusion is a natural choice.
>
> **other network structure**
>
> First, considering that the spatial distribution of stations forms a non-Euclidean structure, CNNs are inherently unsuitable for handling such data. Second, using MLP for fusion introduces two approaches: site-level and global-level. A site-level MLP can only fuse the two types of dynamics latent representations for a given station, without enabling interactions with features from other stations. Conversely, a global-level MLP introduces features from distant stations, which serve as irrelevant noise for the current station. Also, this contrasts with Spatial-MSA, where we aim to let the data-driven model capture features from nearby stations.
>
>
>
> ### **Q2**: Is there other data should be included for geospatial graph construction?
>
> Yes, we utilized SRTM data to construct the GeoSpatial Graph. Naturally, pollutants cannot propagate over large distances or when geographical barriers, such as mountains, exist between stations. The methodology for constructing the GeoSpatial graph is described in detail in Appendix $\underline{\text{A.4 GeoSpatial Graph}}$.

---

> > ### Comment · Reviewer_v3np · 2024-11-24
> > **I keep my current Accept score.**
> >
> > Thanks the authors for the efforts to address my comments.
> > My questions have been clarified and I remain my positive view on this paper. I keep my current Accept score.

---

### Official Review · Reviewer_NWR1 · 2024-11-03

**Soundness:** 3
**Presentation:** 3
**Contribution:** 3
**Rating:** 6
**Confidence:** 3

**Summary:**

This paper introduces Air-DualODE, a novel physics-informed model for predicting air quality that combines the strengths of physics-based and data-driven approaches. Traditional physics-based models are computationally intensive and rely on closed-system assumptions, while data-driven models often lack critical physical insights. Air-DualODE addresses these limitations with a dual-branch design using Neural Ordinary Differential Equations (Neural ODEs): one branch incorporates open-system physical equations to model spatiotemporal dependencies, while the second captures additional dependencies in a data-driven manner. The two branches are temporally synchronized and fused for enhanced predictive accuracy. Experimental results show that Air-DualODE outperforms existing methods in predicting pollutant levels across different spatial areas, making it a robust tool for air quality forecasting.

**Strengths:**

1) Very well written paper combined with appropriate plots.

2) Recognizing that air pollution transport occurs in an open system, the authors redefine the diffusion-advection equation in explicit spaces. This new formulation, termed BA-DAE, aligns the physical equations more accurately with real-world pollutant transport in open air environments, enhancing the model's applicability and reliability.

3) The proposed model, Air-DualODE, uniquely integrates both Physics Dynamics and Data-Driven Dynamics. This dual-branch structure leverages the advantages of physical models (to capture foundational physical behaviors) and data-driven insights (to adapt to complex patterns not covered by physics alone). This approach represents the first dual-dynamics deep learning model specifically tailored for air quality prediction in open systems.

4) Experimental results demonstrate that Air-DualODE outperforms existing models, achieving state-of-the-art accuracy in forecasting pollutant concentrations across diverse spatial scales, from city-wide to national levels.

**Weaknesses:**

The dual-branch structure in Air-DualODE, while innovative, adds complexity to the model, potentially making it less interpretable than simpler models. This may pose challenges for stakeholders, such as policymakers, who require clear explanations of how predictions are made. But this is not a very important point neither does it offer any reason to not accept this paper.

**Questions:**

1) How does Air-DualODE perform in regions with sparse or inconsistent air quality data? Are there mechanisms in place to handle data gaps, or do you recommend a minimum data density for effective predictions?

2) Has Air-DualODE been tested on pollutants other than those mentioned in the paper? Could this framework be adapted to predict other types of environmental data, such as water quality?

---

> ### Author Response · Authors · 2024-11-21
> **Response to Reviewer NWR1**
>
> We would like to sincerely thank Reviewer NWR1 for acknowledging our work's novelty and contributions. Your concerns about sparse data's performance and usage in other domains are resolved in following contents.
>
>
> ### **W1**: stakeholders require clear explanations of how predictions are made. (Although not very important point, we still want to elaborate.)
>
> Thank you for recognizing our work. Actually, our model can provide stakeholders with explanations to some extent. In Section 4.4, the second case study presents a visualization of $\boldsymbol{\beta}$ under two scenarios in Beijing. In our view, $\boldsymbol{\beta}$ can serve as an important reference indicator for stakeholders: $\beta_i < 0$ indicates a sink (e.g., out-of-boundary diffusion/advection or forest absorption around station $i$), while $\beta_i > 0$ indicates a source (e.g., industrial activity or vehicular emissions around station $i$). This provides stakeholders with insights into how the model makes predictions to some degree. We hope this response addresses your concerns.
>
>
> ### **Q1**: Air-DualODE's performance on spare data and minimum data density for effective predictions
>
> **Air-DualODE's performance on spare data**
>
> To study how Air-DualODE performs in regions with sparse or inconsistent air quality data, we vary the size of the training set from 10% to 100%, while keeping the validation and test sets fixed. The results are presented in the table below:
>
> | Beijing | MAE     | RMSE    | SMAPE  |
> | ------- | ------- | ------- | ------ |
> | 10%     | 44.0587 | 67.6437 | 0.8279 |
> | 20%     | 43.5233 | 66.664  | 0.8004 |
> | 30%     | 42.5117 | 65.9448 | 0.7685 |
> | 70%     | 41.7993 | 63.9566 | 0.7672 |
> | ALL     | 40.3208 | 62.0407 | 0.7388 |
>
> | KnowAir | MAE     | RMSE    | SMAPE  |
> | ------- | ------- | ------- | ------ |
> | 10%     | 20.0789 | 31.4796 | 0.4493 |
> | 20%     | 19.8123 | 30.8212 | 0.4454 |
> | 30%     | 19.3563 | 30.5257 | 0.4365 |
> | 70%     | 18.9352 | 29.9471 | 0.429  |
> | ALL     | 18.6431 | 29.3657 | 0.4213 |
>
> The following two conclusions can be drawn from the tables below:
>
> 1. As the amount of training data increases, the performance of Air-DualODE improves correspondingly (scaling law in data sizes).
> 2. Even with only 30% of the training set available, Air-DualODE still performs well and does not degrade significantly.
>
> This demonstrates that our model can still achieve strong performance even with sparse data.
>
> **minimum data density for effective predictions**
>
> According to the experiment results, minimum data density for effective predictions is around 30%.
>
> ### **Q2**: tested on other pollutants and usage in other domain
>
> **tested on other pollutants**
>
> In the experiments, we focused on PM2.5 due to the availability of the dataset (KnowAir only provides three-hour interval data for PM2.5) and to maintain consistency with previous studies. However, we believe that other pollutants, being essentially fine particles, can also be effectively modeled by Air-DualODE.
>
> **usage in other domain**
>
> Thanks for your suggestion. Air-DualODE is specifically designed for air pollutant prediction, and the BA-DAE in its physical branch may not align with the requirements for water quality prediction. Therefore, Air-DualODE is not directly applicable to water quality prediction tasks. However, by revising the Air-DualODE's physical branch with water quality-specific equations, we believe it has the potential to effectively address water quality prediction tasks.

---

> > ### Author Response · Authors · 2024-11-25
> > **Looking forward to your feedback**
> >
> > Dear Reviewer NWR1,
> >
> > Thank you for your valuable and constructive reviews, which has inspired further improvements to our paper. We have made an extensive effort to try to successfully address your concerns, by conducting experiments on sparse data. We hope our response can effectively address your concerns, If you have any further concerns or questions, please do not hesitate to let us know, and we will respond timely.
> >
> > Best regards,
> >
> > Authors

---

### Official Review · Reviewer_6Tsm · 2024-11-04

**Soundness:** 2
**Presentation:** 2
**Contribution:** 2
**Rating:** 6
**Confidence:** 5

**Summary:**

This paper proposes Air-DualODE, a novel approach for air quality prediction that combines physics-based and data-driven methods using dual Neural ODEs. The physics branch implements a modified diffusion-advection equation with a correction term for open systems (BA-DAE). In contrast, the data-driven branch employs masked attention-based Neural ODEs to capture unknown dynamics. The two branches are temporally aligned using a decaying contrastive learning scheme and fused in latent space using GNN, demonstrating superior performance on city-scale (Beijing) and national-scale (KnowAir) datasets.

**Strengths:**

1. The paper addresses the limitations of pure physics-based and pure data-driven approaches by proposing a hybrid framework that attempts to leverage the advantages of both methods.

2. The introduction of BA-DAE with a correction term represents an attempt to model open system dynamics, which is more realistic for air quality prediction than traditional closed system assumptions.

3. The model achieves state-of-the-art performance across different spatial scales while maintaining some level of interpretability through its physics branch.

**Weaknesses:**

1. The paper oversimplifies complex air pollution dynamics using a linear correction term (βX) without proper theoretical justification, undermining its claim of accurate open system modeling.

2. The approach loses physical interpretability when projecting to latent space and violates conservation laws, raising concerns about numerical stability and contradicting the paper's emphasis on physics-informed modeling.

3. The computational efficiency claims are questionable as the dual branch architecture with multiple ODE solvers likely increases computational burden rather than reducing it.

4. The experimental validation is limited, with case studies confined to Beijing data and lacking crucial analyses such as parameter sensitivity testing and solver comparisons.

5. The technical documentation is incomplete, with key mathematical elements missing from figures and insufficient details about architectural choices, making reproducibility challenging.

**Questions:**

Q1. Given that real air pollution sources (industrial activity, vehicle emissions) and sinks (forests, lakes) exhibit complex non-linear relationships, why did you simplify the correction term $\beta X$ as a linear term? What is the physical justification for setting $\beta$'s range to $[−1, +∞)$?

Q2. While you claim that the Physics branch explicitly models physical phenomena, how is this physical interpretability preserved when projecting into latent space?

Q3. Regarding the temporal alignment process using Decay-TCL, how do the chosen values of $\lambda_1 = 1$ and $\lambda_2 =0.8$ guarantee physically meaningful alignment? What is the physical significance of using time-decaying weights?

Q4. Why specifically choose Spatial-MSA in the Data-Driven branch? How does this align with a physics-informed approach?

Q5. The authors justify GNN fusion based on 'distance-dependent influence', but isn't this characteristic already considered in the Physics branch?

Q6. How can the authors justify the performance on the national-scale KnowAir dataset when case studies are limited to the Beijing dataset?

Q7. How does the visualization of $\beta$ values correspond to actual observed pollution source/sink data?"

Q8. Can authors perform sensitivity analysis for different ranges of $\beta$ values?

Q9. The authors seem to only consider the DOPRI5 ODE solver. Could they analyze performance and runtime differences when using simpler methods like Euler or RK4?

Q10. The paper should reference and compare with recent work on climate modeling using diffusion and diffusion-advection equations in neural ODE frameworks [1,2]. Can authors clarify their position by analyzing similarities and differences in their approach to diffusion and advection?

Q11. How does the intentional violation of conservation law in BA-DAE affect numerical stability, particularly for ODE solvers?

Q12. How do you ensure that the BA-DAE in the Physics branch and Neural ODE in the Data-driven branch operate in the same state space?

Q14. Is 'Physics-Informed' appropriate in the title? Would 'Physics-guided' or 'Physics-inspired' be more accurate, given that this might be confused with traditional PINN approaches?

Q15. Can authors provide more details about the RNN used in the Coefficient Estimator?

Q16. Figure 2 lacks several elements mentioned in the text, particularly α from equation 6. The relationship between Dynamics fusion and Section 3.4 equations needs clarification.

Q17. Can authors provide visualizations or distribution analyses showing how Gdiff and Gadv change dynamically with wind speed and direction?

Q18. While criticizing the computational cost of existing physics-based methods, how does your dual branch architecture with a complex fusion mechanism improve efficiency? Doesn't using two ODE solvers increase computational burden?

Q19. Can the authors include the number of forward evaluations (NFE) comparisons in Table 2's ablation studies?

Q20. The authors should cover related work on GNNs that redesign the diffusion equation [3,4] and its variations[5,6] using NODE. Can you discuss more about what the authors' methods have in common and what they differ from?

> [1] Choi, Hwangyong, et al. "Climate modeling with neural advection-diffusion equation." Knowledge and Information Systems 65.6 (2023): 2403-2427.
>
> [2] Hwang, Jeehyun, et al. "Climate modeling with neural diffusion equations." 2021 IEEE International Conference on Data Mining (ICDM). IEEE, 2021.
>
> [3] Wang, Yifei, et al. "Dissecting the diffusion process in linear graph convolutional networks." Advances in Neural Information Processing Systems 34 (2021): 5758-5769.
>
> [4] Chamberlain, Ben, et al. "Grand: Graph neural diffusion." International conference on machine learning. PMLR, 2021.
>
> [5] Thorpe, Matthew, et al. "GRAND++: Graph neural diffusion with a source term." ICLR (2022).
>
> [6] Choi, Jeongwhan, et al. "Gread: Graph neural reaction-diffusion networks." International Conference on Machine Learning. PMLR, 2023.

---

> ### Author Response · Authors · 2024-11-21
> **Response to Reviewer 6Tsm**
>
> We would like to sincerely thank Reviewer 6Tsm for providing a detailed review and insightful comments regarding BA-DAE, hyperparameter's sensitivity, typos and other important components in our model. We have revised our paper accordingly.
>
> ### **W1, Q1**: more discussions about $\boldsymbol{\beta} X$
>
> **1. the reason of linear term.**
>
> From the well-known SIR model, in the infectious equation ($\frac{dI}{dt} = \eta \frac{SI}{N} - \gamma I$) and the recovery equation ($\frac{d R}{dt} = \gamma I$), the growth rate of recovery is modeled as a proportion to infected population. However, in reality, their relationship exhibits complex nonlinear dependencies. Motivated by this design and the principle of Occam’s razor, we also use $\boldsymbol{\beta} X$ to model sources (generation of pollutants) and sinks (dissipation of pollutants). Besides, $\boldsymbol{\beta}$ is estimated by the RNN based coefficient estimator (cf. Fig.2's "Physics Dynamics" part), which accounts for the nonlinear dependencies of sources and sinks.
>
> **2. physical justification for $\boldsymbol{\beta}$'s range.**
>
> As mentioned in Section 3.2.2, the dissipation of pollutants cannot exceed the current pollutant concentration; therefore, $\boldsymbol{\beta}$ has a minimum value of -1, i.e., all pollutants are dissipated. However, the generation of pollutants may far exceed the current concentration. Thus, $\boldsymbol{\beta}$ can theoretically have a maximum value of $+\infty$.
>
>
> ### **W2, Q11**: numerical stability and physical interpretability
>
> **numerical stability**
>
> We did not observe numerical instability during our experiments. Moreover, the training loss curve (now available in the updated version's Appendix $\underline{\text{A.8 Numerical stability}}$) demonstrates that our method converges. Besides, Air-DualODE incorporates normalization techniques, such as Layer Normalization, which contributes to numerical stability.
>
> **physical interpretability**
>
> To clarify, our proposal is not a fully-explainable new physical model but rather a hybrid model or a physics-guided/inspired method (as you mentioned in Q14). The goal of our work is to enable accurate air quality prediction, rather than having fully-explainability. We integrate physical equations into deep learning models, enabling existing physical knowledge to enhance the model’s learning and inference, thus improving prediction accuracy.
>
> ### **W3, Q18**: computational concerns
>
> Our computational efficiency claims are in comparisons with physics-based methods that solve PDEs on a large grid, incurring significant computational costs. While our physical branch also solves the ODEs derived from the Method of Lines (MOL), it is noteworthy that our spatial discretization is observation station-based. This means that the number of ODEs corresponds to the number of observation stations, which effectively reduces the computational time. We theoretically provide a comparison of computational complexity as follows.
>
> |                       | theoretical analysis of time cost                            |
> | --------------------- | ------------------------------------------------------------ |
> | Air-DualODE           | $O(\text{NFE}^{\text{P}} \cdot \| E \| + \text{NFE}^{\text{D}} \cdot N^2 \cdot C + \| E \|)$ |
> | Physics-based methods | $O(T_{grid} \cdot N_{grid}^{\frac{3}{2}})$                   |
>
>
> Here, $\text{NFE}^{\text{P}}$ and $\text{NFE}^{\text{D}}$ refer to the Number of Forward Evaluations for the physics and data-driven branches, respectively. $|E|$ is the number of edge in $G$. $C$ represents the dimension of latent variables. $N$ represents number of neighboring stations around the current station, while $N_{\text{grid}}$ and $T_{\text{grid}}$ denote the grid points in PDE solving. In practical scenarios,  the number of grid points is much larger than the number of stations: $N_{\text{grid}} \gg N$ and the number of grid time step is also greater than NFE: $T_{\text{grid}} \gg \text{NFE}$. For example, on the Beijing dataset with 35 observation stations, predicting 24 steps means $N \leq 35$ and $\text{NFE} \geq 24$ (depending on ODE-solver). In contrast, physics-based methods would require $N_{\text{grid}} \approx 1000,000  ( 1000^2 )$ and $T_{\text{grid}} \approx 1,000$ to keep accuracy. Therefore, the computational cost of Air-DualODE is lower than that of physics-based methods.

---

> ### Author Response · Authors · 2024-11-21
> **Response to Reviewer 6Tsm**
>
> ### **W4, Q6, Q9:** Case studies and hyperparameter, solver sensitivity analysis
>
> **Case studies about KnowAir**
>
> In Table 1, we present comprehensive experiments to demonstrate that our model performs effectively even on a larger and coarser-grained air pollution dataset (KnowAir). Additionally, we have provided visualizations of several case studies on the KnowAir dataset in the updated version's Appendix $\underline{\text{A.9 The case study of KnowAir}}$.
>
> **hyperparameter and solver sensitive analysis**
>
> These sensitivity analyses are provided in the updated version's Appendix $\underline{\text{A.11 Hyperparameter sensitive analysis}}$ and $\underline{\text{A.12 ODE solver sensitive analysis}}$. From the experiments, it is evident that Air-DualODE demonstrates robustness to hyperparameters and solver types. At the ODE solver level, we recommend the adaptive Dopri5 algorithm. Although its computational cost is higher than the other two solvers, it also delivers greater accuracy.
>
>
> ### **W5**: incomplete technical documentation and concerns on reproducibility.
>
> To ensure reproducibility, we have made our code public available, where the link to the code is in the original submission paper's abstract (cf. abstract line 027). We also revised the mathematical part to provide clear access to the details of our architecture.
>
>
>
> ### **Q2**: how to preserve the physical interpretability when projecting into latent space
>
> Our primary goal is to accurately predict pollutants (e.g., PM2.5) rather than to develop a fully interpretable model. Our approach remains within the domain of deep learning. We applied physical modeling to the inputs (PM2.5) and claim that the physical branch explicitly models physical phenomena. However, after projecting into the latent space, the latent variables retain some physical information but do not preserve full physical interpretability.
>
> ### **Q3**: the chosen values of $\lambda_1$ and $\lambda_2$ in Decay-TCL
>
> Both $\lambda_1$ and $\lambda_2$ are used to constrain the latent representations of the two dynamics according to the third rule mentioned in Section 3.4: representations from the same dynamics should be more similar than those from different dynamics. To meet this condition, we must ensure $\lambda_1 > \lambda_2$. A grid search is performed to determine the appropriate values for $\lambda_1$ and $\lambda_2$ while enforcing $\lambda_1 > \lambda_2$.
>
>
> ### **Q4**: the reason of choosing Spatial-MSA and aligning with a physics-informed approach
>
> **the reason of choosing Spatial-MSA**
>
> To capture spatial correlations across different time steps within the dynamic system, a masked self-attention mechanism is incorporated into $\mathbf{F^D}$ as we described in Section 3.3. Specifically, the adjacency matrix of $G$ is employed as a mask in the Spatial-MSA. Intuitively, if no potential transport pathway exists between two stations, their representations should not be correlated. This approach not only improves computational efficiency but also enables the model to focus on relevant information, thereby enhancing its effectiveness across various spatial granularities in prediction scenarios because every station just attends to their nearby, accessible stations.
>
> **aligning with a physics-informed approach**
>
> We could say this is physics-guided because it leverages geospatial intuition (First Law of Geography: Everything is related to everything else, but near things are more related to each other.) rather than explicit physical equations.
>
>
> ### **Q5**: the difference between GNN fusion and Physics branch
>
> Although both branches consider 'distance-dependent influence', they have essentially two distinct targets. In the physics branch, $G_{\text{diff}}$ and $G_{\text{adv}}$ are used exclusively by the GNN to approximate the Laplacian operator. However, during the fusion process, our goal is to account for the interactive influence of the two types of dynamics representations across different nodes. The degree of this interaction also needs to be defined based on spatial distance. To achieve better fusion of the dynamics representations, we again utilize a graph structure based on spatial distances. Both designs are essential to the Air-DualODE framework.

---

> ### Author Response · Authors · 2024-11-21
> **Response to Reviewer 6Tsm**
>
> ### **Q7**: the relationship between visualized $\boldsymbol{\beta}$ and pollution source/sink.
>
> In the case study presented in Section 4.4, we use two specific time points to illustrate the source/sink scenarios in Beijing (dissipation out of boundary and generation from the boundary can also be seen as special sink and source). When $\beta_i > 0$, it indicates that the pollutant growth rate at the station, beyond diffusion and advection, is positive, signifying the presence of a source phenomenon. Conversely, when $\beta_i < 0$ , it represents a sink phenomenon.
>
> In the left part of Fig. 7, there is a northwest wind, leading to an advection out-of-boundary effect at the stations highlighted by the red circles. This manifests as a sink phenomenon, causing their $\beta$ to trend negative. Conversely, some stations exhibit positive $\beta$, which can be attributed to pollutants transported by the wind from outside the boundary to these stations, representing a source phenomenon.
>
> In the right part of Fig. 7, the wind effect is minimal. For boundary stations, diffusion out of the boundary acts as a sink phenomenon. Meanwhile, for internal stations, the morning rush hour in the city contributes to a source phenomenon due to vehicle emissions. In general, the source/sink nature at different stations can help validate the signs of their $\beta$. For instance, stations located in areas with abundant trees often experience sink phenomena.
>
> ### **Q8**: the sensitivity analysis of $\boldsymbol{\beta}$
>
> $\boldsymbol{\beta}$ is not a hyperparameter, so sensitivity analysis cannot be performed. Rather, it is a value estimated from historical data $( X_i, A_i )_{i = 1}^T$ by the coefficient estimator, which is a GRU based RNN, as shown in the Fig.2's "Physics Dynamics" part.
>
>
> ### **Q10**: comparison with climate model and the difference between Air-DualODE and \[1\]\[2\]
>
> **comparison with climate model**
>
> We sincerely appreciate you sharing these two articles. [1, 2] primarily focus on climate forecasting scenarios, introducing the Neural Diffusion Equation (NDE) and Neural Diffusion-Advection Equation (NADE). In contrast, Air-DualODE is specifically designed for spatiotemporal sequence modeling of air pollutant propagation, whereas NDE and NADE target climate prediction tasks. Due to concerns about the differing domains of application, we did not include them as baselines for comparison. However, as both NDE and NADE fall under the category of physics-guided machine learning, we have incorporated them into the related work section in the updated version to provide a more comprehensive review of relevant studies.
>
> **the difference between Air-DualODE and [1, 2]**
>
> NADE maps the current values to a latent space through an encoder, then iteratively solves the diffusion-advection equation to obtain future latent states, and finally gets predictions using a decoder. Additionally, NADE models uncertainty, which is crucial in the field of climate forecasting. Both NADE and ClimODE(Verma et al., 2024) are pioneering works in physics-guided machine learning and are inspiring.
>
> In terms of similarity, both NADE and Air-DualODE utilize the diffusion-advection equation, derived from the mass conservation law, as it plays a vital role even in meteorological domains. However, there are significant differences between NADE and Air-DualODE:
>
> 1. **Modeling Objects**: In Air-DualODE, the diffusion-advection equation primarily models pollutant propagation, while in NADE, it is used to model latent variables in meteorology.
> 2. **Different Equation**: Air-DualODE’s BA-DAE considers an open system, while NADE’s diffusion-advection equation, adhering to the mass conservation law, primarily applies in a closed system.
> 3. **Discretization Method**: Air-DualODE discretizes the equation into ODEs using the Method of Lines (MOL) and approximates the Laplacian operator with GNNs and Laplacian matrices. In contrast, NADE directly replaces the Laplacian operator with a precomputed Laplacian matrix.
> 4. **Physics-Inspired Approach**: Air-DualODE applies the equation explicitly to pollutants in a physical space and combines it with a data-driven NODE for the final results. On the other hand, NADE applies the equation to latent variables and derives the final predictions via a decoder.
>
> ### **Q12**: concerns of two dynamics' latent representations in the different state space
>
> That’s an excellent question. For this very reason, we introduce Decay-TCL, incorporating a decaying weight $w(t, s)$ based on timestamp differences and dynamics types, to regulate the similarity between latent representations. This approach aims to map the two representations into a shared latent space as much as possible, facilitating subsequent fusion.

---

> ### Author Response · Authors · 2024-11-21
> **Response to Reviewer 6Tsm**
>
> ### **Q14**: the suggestion of appropriate title
>
> Thank for your suggestion. We believe you are correct, and we have revised our phrasing in the updated version to avoid potential misunderstandings. We plan to change the title to "**Air Quality Prediction with Physics-Guided Dual Neural ODEs in Open Systems**", if the paper is accepted and the chair approves this title.
>
>
> ### **Q15**: the details of RNN Coefficient Estimator
>
> This is a sequence modeling approach for estimating the coefficients $\boldsymbol{k}, \boldsymbol{\beta} \ \in R^{N \times 1}$. In the implementation, we use a GRU block to construct the RNN, enabling the processing of historical data $(X_i, A_i)_{i = 1}^T$.
>
>
> ### **Q16**: information supplementation in Fig. 2 and the relationship between Dynamics Fusion and Section 3.4
>
> In Eq. (6), $\alpha$ represents a weighted sum of diffusion and advection. This information has been supplemented in Fig. 2 in the updated version. The Dynamics Fusion in Fig. 2 illustrates the inequalities that must be satisfied between the two dynamics latent representations. These relationships are enforced by the Decay-TCL loss, as discussed in Section 3.4.
>
> ### **Q17**: the visualization of $G_{\text{diff}}$ and $G_{\text{adv}}$ change dynamically
>
> It is important to clarify that  $G_{\text{diff}}$ is not dynamically changing, as described in Section 3.2.1 Diffusion Graph. As for $G_{\text{adv}}$, we have provided additional visualizations in updated version's Appendix $\underline{\text{A.13 Visualization of Advection graph}}$.
>
> ### **Q19**: NFE comparisons in Table 2's ablation studies
>
> The following table contains the NFE comparison of Table 2.
>
> | Model                    | Beijing |       |       |         | KnowAir |       |       |         |
> | ------------------------ | ------- | ----- | ----- | ------- | ------- | ----- | ----- | ------- |
> |                          | MAE     | RMSE  | SMAPE | NFE     | MAE     | RMSE  | SMAPE | NFE     |
> | w/o Physics Dynamics     | 42.75   | 63.45 | 0.75  | 14.00      | 19.69   | 30.65 | 0.45  | 14.00      |
> | w/o Data-Driven Dynamics | 44.33   | 65.6  | 0.82  | 20.00      | 21.21   | 33.09 | 0.56  | 19.73 |
> | Explicit Fusion          | 41.32   | 63.08 | 0.76  | 29.27 | 19.06   | 30.28 | 0.43  | 30.40    |
> | Cross-Space Fusion       | 41.97   | 63.12 | 0.77  | 33.64 | 19.27   | 30.54 | 0.44  | 36.00      |
> | w/o Decay-TCL            | 42.34   | 66.56 | 0.82  | 30.73 | 19.10    | 29.79 | 0.43  | 33.73 |
> | w/o Spatial-MSA          | 40.52   | 62.49 | 0.75  | 32.91 | 18.97   | 30.01 | 0.43  | 34.00      |
> | Air-DualODE              | 40.32   | 62.04 | 0.74  | 32.64 | 18.64   | 29.37 | 0.42  | 35.46 |
>
> From the table above, we can draw the following conclusions:
>
> 1. Dual branches result in a slightly higher NFE but deliver better performance compared to a single branch, owing to the use of two ODE solvers.
> 2. Cross-Space Fusion increases the NFE because of the mismatch between explicit physical equations and latent data-driven representations. Therefore, we use Decay-TCL to align the representations in the same space, which reduces both the errors and the NFE, suggesting that it is effective.
>
> ### **Q20**: coverage of advanced GNNs in Related Work and their similarities and differences with Air-DualODE
>
> **coverage of advanced GNNs in Related Work**
>
> Thank you for providing these articles. We believe that some of the GNN designs referenced in these works draw inspiration from the diffusion equation in PDEs. Therefore, we have included them in the related work section of our updated version.
>
> **their differences with Air-DualODE**
>
> In [3, 4, 5, 6], the diffusion equation is used to model different objects compared to Air-DualODE. In these GNNs, the diffusion equation serves as a motivation for improving classical GNNs, primarily addressing challenges such as over-smoothing and gradient vanishing. In contrast, Air-DualODE’s diffusion refers to the physical phenomenon of pollutant transport caused by uneven concentration distributions. In Air-DualODE, we use the diffusion equation to model pollutant transport's dynamics. In summary, the use of diffusion equation in [3, 4, 5, 6] versus in our Air-DualODE are largely orthogonal.

---

> ### Comment · Reviewer_6Tsm · 2024-11-23
>
> Thank you for your responses. While most of your answers address my questions and concerns, I still have remaining questions:
>
> **Regarding reviewer's response to "W2 & Q11" and the newly added Fig. 8**
> 1. I'm not fully convinced that training loss convergence alone can completely prove numerical stability.
> 2. You mentioned that Layer Normalization contributes to numerical stability -- could you provide additional ablation studies on Layer Normalization to support this claim?
>
> **Regarding sudden changes prediction**
>
> 3. How accurate are your predictions when visualized for sudden changes? Specifically, if there's an upward trend in a sudden change period, does your model correctly predict this upward movement?
> 4. Since you're forecasting 24 steps, could you provide each step error comparison between your model and representative baselines? Currently, Table 4 only shows errors by period.
>
> **Regarding NFE analysis**
>
> 5. How does NFE change in each branch of Air-DualODE's dual branch structure?
> 6. You claimed that Decay-TCL "effectively reduces NFE" -- could you explain more about what you mean by "effectively reduces NFE" in your result?
> 7. How do you explain that removing Spatial-MSA slightly decreases NFE?
>
> **Regarding GNN implementation**
>
> 8. I've reviewed your GNN layer sensitivity study. There appears to be notation confusion between '$n$' in the main paper and '$n$' in Appendix A.6. Your sensitivity analysis shows little variation from 1 to 5 layers. Could you show results for both datasets when using no GNN layer ($n=0$), i.e., using just $Z_t^{(0)}$ as mentioned in line 354? *I will carefully review how this response aligns with and makes sense to the authors' claim to reviewer `v3np` that "GNN Fusion is a natural choice".*

---

> ### Author Response · Authors · 2024-11-24
> **Response to Reviewer 6Tsm**
>
> We are glad that our responses resolve most of your questions and concerns from the original review. Your new questions are addressed as follows.
>
>
> ### **Q1**: I'm not fully convinced that training loss convergence alone can completely prove numerical stability.
>
> We have not observed any numerical instability issues in our experiments. If you have more specific suggestions on investigating numerical stability, we are happy to provide more information according to your suggestions.
>
> ### **Q2**: relationship between layer normalization and numerical stability
>
> When removing all layer normalization from Air-DualODE, gradient explosion and `NAN` occurred after a few epochs, preventing us to have a stable model.
>
>
> ### **Q3**: sudden change visualization
>
> We provide a visualization of the results for sudden changes in Appendix $\underline{\text{A.15 Visualization of sudden changes' results}}$ (Fig.12). The visualizations demonstrate that Air-DualODE can effectively predict upward and downward trends during periods of sudden change.
>
> ### **Q4**: each step error comparison
>
> Considering that a comparison of errors for all 24 steps would be overwhelming, we have provided the error comparison for the 1st, 3rd, 5th, …, 21st, and 23rd steps in Appendix $\underline{\text{A.10 More experiment results}}$ (Table 5) of the updated version.
>
>
> ### **Q5**: How does NFE change in each branch of Air-DualODE's dual branch structure?
>
> NFE changes in each branch depend on their structures because, similar to solving ODEs, different ODEs require different iteration steps (NFE) to achieve numerical precision. From the table in Q19, Physics dynamics' NFE is larger than Data-Driven dynamics' NFE.
>
> ### **Q6**: You claimed that Decay-TCL "effectively reduces NFE" -- could you explain more about what you mean by "effectively reduces NFE" in your result?
>
> "Effectively reduces NFE" refers to the advantage of `Air-DualODE` over `Cross-Space Fusion`. In `Cross-Space Fusion`, the mismatch between the explicit physical equation and the latent data-driven representations leads to large NFE. By placing both of them into the latent space and aligning them using Decay-TCL, `Air-DualODE` achieves smaller NFE, thus effectively reducing NFE compared to `Cross-Space Fusion`.
>
> ### **Q7**: removing Spatial-MSA slightly decreases NFE
>
> Because Spatial-MSA prevents the Data-Driven branch from focusing on distant stations, the adaptive ODE solver needs to extract more useful representations from fewer stations within the Data-Driven branch. As a result, Spatial-MSA slightly increases the NFE of Air-DualODE.
>
>
> ### **Q8**: notation confusion and ablation study of GNN Fusion
>
> **notation confusion**
>
> We revise this notation confusion in the updated version.
>
> **ablation study of GNN Fusion**
>
> We could provide the results for both datasets when no GNN layers are used. To enable fusion across stations, we replaced the GNN Fusion with an MLP structure, which maps $\mathbb{R}^{N \times 2D} \to \mathbb{R}^{N \times 1}$ for pollutant prediction. The results are as follows.
>
> | Model          | Beijing |       |       | KnowAir |        |       |
> | -------------- | ------- | ----- | ----- | ------- | ------ | ----- |
> |                | MAE     | RMSE  | SMAPE | MAE     | RMSE   | SMAPE |
> | w/o GNN Fusion | 41.49   | 63.14 | 0.76  | 21.5    | 32.919 | 0.56  |
> | Air-DualODE    | 40.32   | 62.04 | 0.74  | 18.64   | 29.37  | 0.42  |
>
> It can be observed that w/o GNN Fusion results in a performance drop compared to Air-DualODE on both datasets. This drop is particularly significant on the larger, nationwide KnowAir dataset. Due to the large distances between some stations, using an MLP structure for fusion introduces representations from distant stations, which act as noise for predicting pollutant concentrations at the current station. Consequently, the degradation is more obvious on larger KnowAir dataset.

---

> ### Author Response · Authors · 2024-11-24
> **Request for Reconsideration of the score**
>
> Dear Reviewer 6Tsm,
>
> We hope that we have clarified the new questions you have raised. If we have satisfactorily addressed all your questions, we kindly ask you to reconsider your score.
>
> Thanks for your valuable suggestions and comments to improve our work !
>
> Best Regards,
>
> Authors

---

> ### Comment · Reviewer_6Tsm · 2024-11-24
>
> Thank you for your response. I have remaining questions for clarification.
>
> ---
>
> **Questions about model stability and implementation**
>
> **Q1.** Regarding the observed behavior when removing layer normalization, I'm curious about the distinction between training instability and numerical stability. Additionally, could the NaN occurrences be related to the Dopri5's relative and absolute tolerance settings to 1e-3? Have alternative tolerance values been explored?
>
> **Q2.** The authors claim that Decay-TCL "effectively reduces" NFE. However, the numerical results show relatively small differences. Could you elaborate on what constitutes an "effective reduction" in this context?
>
> **Q3.** About the ablation study of GNN Fusion, would it be possible to provide experimental results comparing equivalent numbers of GNN layers and mapping (i.e., MLP) layers? Additionally, could RMSE and SMAPE metrics be included alongside the MAE values reported in Figure 10?
>
> **Q4.** For the mapping by MLP from $\mathbb{R}^{N\times 2D} \rightarrow \mathbb{R}^{N\times 1}$, have alternative approaches such as mean pooling or sum pooling been considered alongside the MLP implementation? Could the authors further justify the necessity of GNN by comparing the results using these pooling methods?
>
> **Questions about sudden changes prediction**
>
> *The visualization of sudden changes prediction raises several questions about the model's capabilities:*
>
> **Q5**. In Figure 12, there appear to be challenges in predicting significant changes: Figures 12(a) and 12(b) show limitations in capturing sharp increases during time steps 20-30, while (c) and (d) show difficulties in representing post-decrease volatility. Could you clarify how these results demonstrate effective handling of trend changes?
>
> **Q6.** The predictions seem to follow smoothed trends rather than capturing sudden changes. Could you explain more about how this aligns with the goal of sudden change prediction?
>
> **Q7.** Would it be possible to provide more comprehensive quantitative metrics for sudden change prediction?

---

> ### Author Response · Authors · 2024-11-25
> **Response to Reviewer 6Tsm**
>
> ## Questions about model stability and implementation
>
> ### **Q1.** exploration on alternative tolerance
>
> We experimented with alternative tolerance, such as $10^{-2}$ and $10^{-5}$, but the issue of `NaN` still occurs. We believe that layer normalization plays an important role in stabilizing the training process.
>
> ### **Q2.** The authors claim that Decay-TCL "effectively reduces" NFE. However, the numerical results show relatively small differences. Could you elaborate on what constitutes an "effective reduction" in this context?
>
> Sorry for the confusion. Compared to `Cross-Space Fusion` as shown in the table w.r.t Q19 in the original review, `Air-DualODE` (which uses Decay-TCL) reduces both the three error metrics and the NFE, which is highly desirable. Thus, we think that Decay-TCL is effective.
>
> To avoid confusion, we rephrase that sentence as follows: Therefore, we use Decay-TCL to align the representations in the same space, which reduces both the errors and the NFE, suggesting that it is effective.
>
>
> ### **Q3, Q4:** comparison with equivalent numbers of MLP layers, more metrics in Fig.10 and more ablation studies about GNN Fusion
>
> **comparison with equivalent numbers of MLP layers**
>
> We provide the experimental results for the equivalent MLP layers. The results are as follows:
>
> | Model                                | Beijing |       |       | KnowAir |        |       |
> | ------------------------------------ | ------- | ----- | ----- | ------- | ------ | ----- |
> |                                      | MAE     | RMSE  | SMAPE | MAE     | RMSE   | SMAPE |
> | MLP Fusion (one layer)               | 41.49   | 63.14 | 0.76  | 21.50    | 32.919 | 0.56  |
> | Equivalent MLP Fusion (three layers) | 41.45   | 63.23 | 0.75  | 21.42   | 32.83  | 0.56  |
> | Mean-Pooling Fusion                  | 43.23   | 68.88 | 0.80  | 20.03   | 31.22  | 0.45  |
> | Sum-Pooling Fusion                   | 43.12   | 68.30 | 0.80  | 19.97   | 31.10  | 0.44  |
> | Air-DualODE                          | 40.32   | 62.04 | 0.74  | 18.64   | 29.37  | 0.42  |
>
> **more metrics in Figure 10**
>
> Following your suggestions, we now include the RMSE and SMAPE metrics to prove the Air-DualODE's robustness in Fig.10.
>
> **more ablation study about GNN Fusion**
>
> We also provide the results of mean pooling fusion and sum pooling fusion in the above table. The results suggest that Pooling Fusion performs worse than MLP Fusion, and GNN Fusion. Therefore, this justify the necessity of using GNN.
>
>
> ## Questions about sudden changes prediction
>
> ### **Q5, Q6, Q7**: sudden changes
>
> To clarify, we include the experimental setting of sudden changes is mainly to ensure empirical fairness and consistency with the two SOTA methods --- both AirPhyNet and Airformer have this setting in their experiments. In our proposal, we do not have a specific design goal to capture sudden changes, neither does AirPhyNet nor Airformer. From both the aggregated results in Table1 in the original paper and the step-wise metrics shown in Table5 to respond your question Q4 in the second round review, it all demonstrates that Air-DualODE outperforms AirPhyNet and Airformer under sudden changes, though we cannot guarantee that our proposal can well capture every sudden change. We leave specific designs on better handling sudden changes as a promising future research direction.

---

> > ### Author Response · Authors · 2024-11-25
> > **Contribution Clarification**
> >
> > We would like to clarify that the paper's contribution mainly lie in open system modeling and the DualODE design to model both physical and data-driven  dynamics, which is acknowledged by you in the original review. In addition, we hope that you appreciate our efforts on elaborating the technical details and that you are satisfied with our responses. If so, we kindly ask you to reconsider your score.

---

> ### Comment · Reviewer_6Tsm · 2024-11-25
>
> Thank you for your detailed responses. I have several suggestions that I believe would strengthen the paper.
>
> First, I notice that both Sum-Pooling and Mean-Pooling actually outperform MLP Fusion on the KnowAir dataset. Could you update the paper to provide a more nuanced justification for the "necessity of using GNN" through a comprehensive comparison of these results? This would make your argument more convincing.
>
> Regarding layer normalization and stability, I suggest addressing the dependency on layer normalization as a limitation section. Specifically, the fact that the model cannot work without layer normalization due to NaN values is an important limitation that should be explicitly discussed.
>
> While I understand that sudden change prediction is not the primary claim of your paper, I believe the current presentation could be improved. Instead of claiming to "effectively handle" sudden changes, I suggest providing a more balanced analysis of Figure 12, explicitly discussing both successful and unsuccessful predictions. This could include acknowledging the limitations in capturing sudden changes and reframing this analysis as an exploration of the model's capabilities rather than a definitive solution.
>
> I recommend expanding the limitations and future work sections to include a clear discussion of the model's current limitations in handling sudden changes, the dependency on layer normalization and its implications, and potential future research directions for addressing these limitations.
>
> Based on how these suggestions are incorporated into the paper, I would be willing to reconsider my evaluation.

---

> > ### Author Response · Authors · 2024-11-27
> > **Response to Reviewer 6Tsm**
> >
> > Many thanks for your constructive suggestions to improve our manuscript. We have now revised our paper in the updated version. The summary of changes is as follows:
> >
> > 1. We revise the explanation of GNN Fusion (cf. lines 351-355) and add an effective analysis of GNN Fusion in the $\underline{\text{Appendix A.16 Ablation study on GNN Fusion}}$.
> > 2. We update the content of $\underline{\text{Appendix A.15 Visualization of sudden changes' results}}$ to provide a more balanced analysis of Figure 12.
> > 3. We include $\underline{\text{Appendix A.17 Limitations Discussion}}$ to discuss Air-DualODE’s current limitations more clearly, and future work is included in $\underline{\text{Section 5: Conclusion and Future Work}}$.
> >
> > Despite these limitations, we believe that we have made considerable contributions on modeling open systems and designing a hybrid DualODE framework, and these limitations do not affect these contributions.

---

> ### Comment · Reviewer_6Tsm · 2024-11-28
>
> Thank you for your thorough review of the revised paper. I have several follow-up questions and concerns regarding your modifications:
>
> ---
>
> I appreciate the analysis in `Section A.16`, which helps justify the use of GNN. However, there seems to be a contradiction in your explanations:
>
> **Q1.** For KnowAir, `Figure 10` shows minimal change in error metrics as the number of layers increases, which appears inconsistent with your explanation about performance degradation due to incorporating distant node representations (`lines 1046-1048`). Could you clarify at which point the layers begin to show degradation in `Figure 10` via an additional sensitivity study?
>
> **Q2.** Additionally, mean pooling incorporates representations from all nodes, including distant ones, yet performs better than MLP. This seems to contradict your explanation about distant node representations acting as noise. Could you reconcile these apparent contradictions?
>
>
> ---
>
> While your explanation about layer normalization dependency is reasonable when using Dopri5, I have two questions:
>
> **Q3.** Have you tested whether NaN values occur when using simpler solvers like RK4 or Euler method without layer normalization?
>
> **Q4.** How can you claim that layer normalization dependency doesn't affect the model's practicality and scalability, especially when the model fails without it? This seems to contradict your statement that "these limitations do not affect these contributions." Could you refine and expand on this?
>
> **Q5.** While `A.17` is a good addition, I suggest expanding it to include limitations identified via discussions with other reviewers, discussing broader challenges in open system modeling and hybrid DualODE framework design, and providing more concrete future research directions for addressing these limitations.
>
> ---
>
> **Q6.** Your current evaluation metrics (MAE, RMSE, MAPE) may not be the most appropriate for evaluating sudden change predictions. I suggest below aspects:
> The authors need to consider including [mean directional accuracy (MDA)](https://en.wikipedia.org/wiki/Mean_directional_accuracy) to evaluate prediction direction (up/down) accuracy [2,3] and consider change point detection evaluation metrics [1]. Either the authors supplement your experiments with these metrics or acknowledge this limitation. The authors must add this discussion in the main text about the challenges of evaluating sudden change predictions.
>
> Given the extended rebuttal period, I encourage you to either:
> a) strengthen your experimental analysis using these suggested metrics or
> b) expand your limitations section to acknowledge these evaluation challenges and discuss potential alternative approaches.
>
> ---
>
> > [1] Van den Burg, Gerrit JJ, and Christopher KI Williams. "An evaluation of change point detection algorithms." arXiv preprint arXiv:2003.06222 (2020).
> >
> > [2] Witzke, Simon, et al. "Mobility data improve forecasting of COVID-19 incidence trends using graph neural networks." epiDAMIK 6.0: The 6th International workshop on Epidemiology meets Data Mining and Knowledge Discovery at KDD 2023. 2023.
> >
> > [3] Blaskowitz, Oliver, and Helmut Herwartz. "On economic evaluation of directional forecasts." International journal of forecasting 27.4 (2011): 1058-1065.

---

> > ### Author Response · Authors · 2024-11-29
> > **Response to Reviewer 6Tsm**
> >
> > Your follow-up questions and concerns are addressed as follows.
> >
> > ### **Q1**: layers begin to show degradation
> >
> > We conduct experiments with additional layers in the GNN Fusion module, and observe that when the number of layers increases to 30, degradation begins to occur. When the number of layers is fewer than 30, no significant degradation is observed. This is because we utilized residual connections between GNN layers. Previous studies have shown that residual connections enable deeper GNN architectures by mitigating issues such as over-smoothing [1, 2].
> >
> > | KnowAir-n | MAE   | RMSE  | SMAPE |
> > | --------- | ----- | ----- | ----- |
> > | 1         | 18.94 | 29.71 | 0.42  |
> > | 3         | 18.64 | 29.37 | 0.42  |
> > | 5         | 18.78 | 29.76 | 0.42  |
> > | 10        | 18.75 | 29.74 | 0.42  |
> > | 15        | 18.88 | 29.98 | 0.42  |
> > | 20        | 19.04 | 30.05 | 0.43  |
> > | 30        | 19.27 | 30.12 | 0.43  |
> > | 40        | 24.81 | 39.80 | 0.55  |
> > | 50        | 25.38 | 71.36 | 0.55  |
> >
> > > [1]. Li G, Muller M, Thabet A, et al. Deepgcns: Can gcns go as deep as cnns?[C]//Proceedings of the IEEE/CVF international conference on computer vision. 2019: 9267-9276.
> > >
> > > [2]. Li G, Müller M, Ghanem B, et al. Training graph neural networks with 1000 layers[C]//International conference on machine learning. PMLR, 2021: 6437-6449.
> >
> >
> > ### **Q2**: clarification of mean pooling fusion
> >
> > To clarify, this process does not incorporate representations from other nodes (cf. lines 1030–1031). Specifically, the dimension of $Z^{(0)}_t$ in GNN Fusion (Section 3.4 Dynamics Fusion) is $\tau \times N \times D$, where $\tau$ represents the prediction length, $N$ denotes the number of stations, and $D$ means the dimensions of the latent representations of each station after concatenation. Considering that the prediction results have the shape of $\tau \times N \times 1$, we apply mean pooling fusion along the feature dimension $D$ ($D$-Mean-Pooling Fusion).  Therefore, it does not contradict our explanation that distant node representations act as noise.
> >
> > To avoid confusion, we also include experiments on mean pooling fusion along the station dimension $N$ of node representations ($N$-Mean-Pooling Fusion. This process incorporates representations from some other nodes. Specifically, these nodes' representations are aggregated based on the order of stations in the representation instead of distances. The results are as follows.
> >
> > | Model                                | Beijing |       |       | KnowAir |       |       |
> > | ------------------------------------ | ------- | ----- | ----- | ------- | ----- | ----- |
> > |                                      | MAE     | RMSE  | SMAPE | MAE     | RMSE  | SMAPE |
> > | MLP Fusion (one layer)               | 41.49   | 63.14 | 0.76  | 21.50   | 32.92 | 0.56  |
> > | Equivalent MLP Fusion (three layers) | 41.45   | 63.23 | 0.75  | 21.42   | 32.83 | 0.56  |
> > | $D$-Mean-Pooling Fusion              | 43.23   | 68.88 | 0.80  | 20.03   | 31.22 | 0.45  |
> > | $N$-Mean-Pooling Fusion              | 41.71   | 64.12 | 0.76  | 20.89   | 32.09 | 0.51  |
> > | Air-DualODE                          | 40.32   | 62.04 | 0.74  | 18.64   | 29.37 | 0.42  |
> >
> > For the Beijing dataset, stations are very close to each other, so incorporating features from other nodes helps improve the final prediction results ( see MLP-based Fusion and $N$-Mean-Pooling Fusion). Among these, the experiments show that GNN Fusion performs the best. Conversely, $D$-Mean-Pooling Fusion, which does not incorporate features from other nodes, performs the worst.
> >
> > For the KnowAir dataset, stations are relatively far from the target station. The significant contrast between MLP-based Fusion and $D$-Mean-Pooling Fusion suggests that incorporating representations from distant nodes introduces noise. Additionally, $N$-Mean-Pooling Fusion also performs poorly because it does not necessarily incorporate representations from adjacent nodes.
> >
> > In summary, these results do not contradict our view.

---

> ### Author Response · Authors · 2024-11-29
> **Response to Reviewer 6Tsm**
>
> ### **Q3**: simpler solvers without layer normalization
>
> We conduct several experiments using simpler fixed-step solvers, such as RK4 and Euler, without Layer Normalization. The results are as follows.
>
> | w/o Layer Normalization | MAE   | RMSE  | SMAPE |
> | ----------------------- | ----- | ----- | ----- |
> | Beijing-euler           | 41.52 | 63.78 | 0.75  |
> | Beijing-rk4             | 41.32 | 63.22 | 0.75  |
> | Beijing-dopri5          | /     | /     | /     |
> | KnowAir-euler           | 18.99 | 30.78 | 0.42  |
> | KnowAir-rk4             | 18.98 | 30.45 | 0.42  |
> | KnowAir-dopri5          | /     | /     | /     |
>
> The results demonstrate that both RK4 and Euler can train stably without Layer Normalization. These empirical findings suggest that the dependency on Layer Normalization is related to whether using fixed-step solvers, as Dopri5 is a ODE solver using adaptive steps. The dual dynamics, with their inconsistent numerical ranges, lead to instability during adaptive step sizes in the forward and backward propagation. However, Layer Normalization ensures consistency of the numerical ranges in the dual dynamics’ adaptive solving during each iteration, thereby ensuring numerical stability.
>
> | w/ Layer Normalization | MAE   | RMSE  | SMAPE |
> | --------------------- | ----- | ----- | ----- |
> | Beijing-euler         | 41.23 | 63.09 | 0.74  |
> | Beijing-rk4           | 40.80 | 62.90 | 0.74  |
> | Beijing-dopri5        | 40.32 | 62.04 | 0.74  |
> | KnowAir-euler         | 18.92 | 30.77 | 0.42  |
> | KnowAir-rk4           | 18.94 | 30.42 | 0.42  |
> | KnowAir-dopri5        | 18.64 | 29.37 | 0.42  |
>
> The above table provides the other simpler solvers' results (with layer normalization). The results demonstrate that w/o Layer Normalization slightly decreases model performance. Additionally, using the adaptive ODE solver Dopri5 yields the best results. To summarize, in settings where layer normalization is not available, we can go for fix-step solvers. Otherwise, we go for Dopri5 with layer normalization, which gives the best accuracy.
>
> ### **Q4**: layer normalization dependency
>
> Layer Normalization is a common component used to constrain numerical ranges in deep learning architectures like Transformers. It is not a new, additional part introduced by our proposal. In addition, as indicated in the previous comment, if there exist settings where layer normalization are unavailable, we can use fix step size based solvers.
>
>
>
> ### **Q5**: including other reviewer's limitations
>
> We include limitations and discussions mentioned by other reviewers in the Appendix A.17 and A.18. However, we have sufficiently addressed all other reviewers' questions, and they all maintain positive outlook on our paper.
>
>
>
> ### **Q6**: sudden changes definition and evaluation
>
> Although the rebuttal period has been extended, the revision submission deadline remains unchanged. From the time we received your questions and concerns to the final submission deadline, the time available was very limited (i.e., around 8 hours). Therefore, we chose your second approach. To clarify, both our sudden change definition and evaluation metrics follow previous works (e.g., Airformer and AirPhyNet) to ensure empirical setting consistency, leading to fair comparisons. We include these evaluation challenges and discuss potential alternative approaches you suggested in both the main text (cf. lines 428-431) and Appendix A.18. In future work, we will consider your suggestions regarding the definition and evaluation of sudden changes.

---

> > ### Author Response · Authors · 2024-11-29
> > **Request for Reconsideration of the score**
> >
> > We truly appreciate your 49 (25+8+7+3+6) insightful and constructive suggestions and questions, which helps us improve our manuscript significantly. If you find that our manuscript, after incorporating all 49 of your valuable feedback points, meets your expectations, we would be truly grateful if you could kindly consider raising your score.

---

> ### Comment · Reviewer_6Tsm · 2024-11-30
>
> Thank you for your dedicated engagement in our discussion. I am pleased to recognize your efforts and proactive improvements by raising my initial `rating` by 2 points and `confidence`, and `contribution` by 1 point each. While I spent considerable time reviewing your responses and aimed to reply promptly, I appreciate your understanding if you think I am late.
>
> Your responses to Q1 and Q2 are satisfied, providing detailed experimental results and clearly distinguishing between $D$-Mean-Pooling Fusion and $N$-Mean-Pooling Fusion, along with well-explained differences between datasets.
>
> Regarding Q3, you have appropriately addressed my concerns with relevant results.
>
> While you note that layer normalization is a common component in various models, I believe its significance in Neural ODEs could benefit from additional interpretation [1]. Based on [1], I recommend expanding your discussion to include:
> First, layer normalization is well-suited for NODEs as it normalizes features rather than batches, aligning with NODEs' parameter sharing across continuous time steps. Second, it demonstrates strong compatibility with NODEs' continuous nature by avoiding batch statistics dependency. Additionally, layer normalization plays a crucial role in stabilizing the learning process by ensuring smoother dynamics of hidden representations, especially given NODEs' sensitivity to normalization technique selection.
>
>
> Given that you've demonstrated functionality without Layer Normalization using fixed-step solvers, I suggest a more detailed discussion in your Appendix. From my experience, Dopri5 solving time in your model will likely be quite substantial. Could you report training times rather than inference times?
>
> Regarding Q6, I find it somewhat unsatisfactory that your paper remains constrained within the conventional boundaries set by previous studies such as Airformer and AirPhyNet. During this extended rebuttal period, I would be delighted to see even a small demonstration of the potential to better these existing limitations.
> Although the authors cannot cover this perfectly due to time constraints, I suggest this because I believe that if the authors can differentiate from existing methods in this aspect, your research value will be more distinct.
>
> > [1] Gusak, Julia, et al. "Towards understanding normalization in neural odes." ICLR 2020.

---

> > ### Author Response · Authors · 2024-12-01
> > **Response to Reviewer 6Tsm**
> >
> > Thank you sincerely for raising your score and for recognizing our efforts on improving the manuscript based on your comments.
> >
> > Layer normalization discussions: Since the last submission deadline has already passed, we are unable to submit an updated version to OpenReview anymore. Therefore, we plan to include additional detailed discussions according to your suggestions in the final version's Appendix.
> >
> > We provide the training times as follows (one epoch).
> >
> > | Training time      | Beijing | KnowAir |
> > | ------------------ | ------- | ------- |
> > | Air-DualODE-Euler  | 53s     | 108s    |
> > | Air-DualODE-RK4    | 68s     | 274s    |
> > | Air-DualODE-Dopri5 | 81s     | 378s    |
> >
> > This result shows that dopri5 requires longer training time, but considering that training is conducted offline, sacrificing training time for improved model accuracy is acceptable. However, in scenarios where training time is a critical factor, Euler or RK4 solvers can be used at the expense of model accuracy. Air-DualODE allows different types of users to select the ODE solver based on their specific requirements. That is a trade-off for different scenarios.
> >
> > Regarding Q6: Based on the requirements mentioned in your original Q6,
> >
> > > I encourage you to either: a) strengthen your experimental analysis using these suggested metrics or b) expand your limitations section to acknowledge these evaluation challenges and discuss potential alternative approaches.
> >
> > We chose to follow your second approach and believe we have addressed this issue according to the suggestions in Q6. You can find the following limitation discussions about sudden changes definitions and evaluations in the main text (cf. lines 428–431).
> >
> > > Discussions: We use the current definition of sudden changes and the corresponding metrics to ensure consistency with existing studies (Liang et al., 2023; Hettige et al., 2024). However, they may not be the most appropriate ones. In future work, we plan to explore alternative evaluation metrics, such as mean directional accuracy (MDA) (Van den Burg & Williams, 2020), and adopt new sudden change definitions based on change point detection algorithms (Witzke et al., 2023).
> >
> > We truly appreciate that you raise the score, but we would also like to understand if there are any remaining concerns preventing you from giving us an acceptance score. From your most recent reply, we feel that the only unresolved concern might be our decision not to follow your previously suggested first approach (`strengthen your experimental analysis using these suggested metrics`). We would like to confirm if this is the key reason for your score being below the acceptance bar. Although the remaining rebuttal period is limited, we are now trying our best to provide a small demonstration experiment of sudden changes’ detection algorithms and evaluation metrics. We hope to be able to offer this experiment before the end of the discussion phase.

---

> > > ### Author Response · Authors · 2024-12-02
> > > **Response to Reviewer 6Tsm**
> > >
> > > > Given the extended rebuttal period, I encourage you to either: a) strengthen your experimental analysis using these suggested metrics or b) expand your limitations section to acknowledge these evaluation challenges and discuss potential alternative approaches.
> > >
> > > Regarding Q6: To follow your suggestion on conducting additional experiments on sudden changes (approach (a)), we use two new sudden change definitions based on two different change point detection algorithms (Binary Segmentation [1] and Kernel Change Point Detection [2]) and use the new evaluation metric `MDA`, as detailed below.
> > >
> > > `Table1` The original sudden change definition (cf. lines 399 to 416):
> > >
> > > |             | Beijing   |           |          |          | KnowAir   |           |          |          |
> > > | ----------- | --------- | --------- | -------- | -------- | --------- | --------- | -------- | -------- |
> > > |             | MAE       | RMSE      | SMAPE    | MDA      | MAE       | RMSE      | SMAPE    | MDA      |
> > > | Airformer   | 68.80     | 91.16     | 0.75     | 0.53     | 39.99     | 55.35     | **0.49** | 0.55     |
> > > | AirPhyNet   | 70.03     | 94.60     | 0.78     | 0.51     | 43.23     | 58.79     | 0.50     | 0.51     |
> > > | Air-DualODE | **66.40** | **90.31** | **0.73** | **0.57** | **39.79** | **54.61** | **0.49** | **0.57** |
> > >
> > > `Table2` The new sudden change definition based on the **Binary Segmentation detection**:
> > >
> > > |             | Beijing   |           |          |          | KnowAir   |           |          |          |
> > > | ----------- | --------- | --------- | -------- | -------- | --------- | --------- | -------- | -------- |
> > > |             | MAE       | RMSE      | SMAPE    | MDA      | MAE       | RMSE      | SMAPE    | MDA      |
> > > | Airformer   | 45.23     | 65.58     | 0.78     | 0.40     | 20.59     | 32.25     | **0.45** | **0.44** |
> > > | AirPhyNet   | 46.63     | 67.71     | 0.80     | 0.38     | 23.00     | 33.65     | 0.50     | 0.34     |
> > > | Air-DualODE | **43.47** | **65.35** | **0.76** | **0.45** | **20.13** | **31.56** | **0.45** | **0.44** |
> > >
> > > `Table3` The new sudden change definition based on the **Kernel Change Point Detection**:
> > >
> > > |             | Beijing   |           |          |          | KnowAir   |           |          |          |
> > > | ----------- | --------- | --------- | -------- | -------- | --------- | --------- | -------- | -------- |
> > > |             | MAE       | RMSE      | SMAPE    | MDA      | MAE       | RMSE      | SMAPE    | MDA      |
> > > | Airformer   | 45.43     | 67.33     | 0.74     | **0.46** | 20.50     | 32.09     | 0.45     | **0.42** |
> > > | AirPhyNet   | 47.27     | 69.75     | 0.76     | 0.39     | 22.87     | 33.53     | 0.49     | 0.36     |
> > > | Air-DualODE | **43.92** | **66.92** | **0.72** | **0.46** | **20.04** | **31.41** | **0.44** | **0.42** |
> > >
> > > **Note:** In these two new sudden change definitions, we do not impose the constraint of PM2.5 being greater than or equal to 75 when using the algorithm to identify change points. Therefore, the value ranges differ from those in the original definition.
> > >
> > > From the above three tables, Air-DualODE is indeed better than the baseline models under different definitions of sudden changes and a variety of evaluation metrics.
> > >
> > > > [1]. Scott, Andrew Jhon, and Martin Knott. "A cluster analysis method for grouping means in the analysis of variance." *Biometrics* (1974): 507-512.
> > > >
> > > > [2]. Harchaoui, Zaid, Eric Moulines, and Francis Bach. "Kernel change-point analysis." *Advances in neural information processing systems* 21 (2008).

---

> ### Comment · Reviewer_6Tsm · 2024-12-02
>
> Thank you for your response. I value your dedication to addressing the limitation I pointed out about sudden changes by incorporating 2 new change point detection approaches and evaluating them using the MDA metric.
>
> In line 375, you define sudden changes as "cases where PM2.5 levels exceed 75 μg/m³ and fluctuate by more than ±20 μg/m³ in the next three hours," but in your note, you mention, "we do not impose the constraint of PM2.5 being greater than or equal to 75 when using the algorithm to identify change points."
> *Could you explain how this difference affects the interpretation of results in the three tables?*
>
> The comprehensive analysis provided by the authors confirms that Air-DualODE performs well in terms of sudden changes. *Can the authors further discuss the potential threats to validity that may arise when using these metrics and two kinds of evaluation methods (i.e., binary segmentation and kernel change point detection)?*
>
> ---
>
> Initially, I rated this paper at `3`. Through our extensive discussion and your responses to numerous feedback points, I raised my rating to `5`.
>
> Now, with this response showing your readiness to go beyond the conventional evaluation approaches used by Airformer and AirPhyNet by incorporating more sophisticated change point detection evaluation methods and the MDA metric, I am increasing my rating to `6`. This last improvement demonstrates your commitment to advancing the field beyond existing limitations and provides a more rigorous evaluation framework for future research in this area.

---

> > ### Author Response · Authors · 2024-12-03
> > **Response to Reviewer 6Tsm**
> >
> > Thank you sincerely once again for raising your score and for recognizing our efforts to improve the manuscript based on your comments during the rebuttal period. Your recognition is the greatest encouragement for our work.
> >
> > ### **Q1**: Could you explain how this difference affects the interpretation of results in the three tables?
> >
> > Sorry for the confusion. Considering the new definition of sudden change, we have completely abandoned the original sudden change settings. Specifically, the original sudden change definition only considered cases where $\text{PM2.5} \ge 75$μg/m³. Conversely, the two new sudden change definitions account for the entire range of $\text{PM2.5}$ levels, including cases where $\text{PM2.5} \le 75$μg/m³, which results in the value range appearing smaller compared to the original definition (e.g., in the Beijing dataset, the original definition’s MAE value range is around 67, while the new definition’s MAE value range is around 45). However, for consistency and alignment with the original sudden change’s value range, we provide the change point detection (with the constraint $\text{PM2.5} \ge 75$μg/m³) results as follows.
> >
> > `Table1` The original sudden change definition (cf. lines 399 to 416):
> >
> > |             | Beijing   |           |          |          | KnowAir   |           |          |          |
> > | ----------- | --------- | --------- | -------- | -------- | --------- | --------- | -------- | -------- |
> > |             | MAE       | RMSE      | SMAPE    | MDA      | MAE       | RMSE      | SMAPE    | MDA      |
> > | Airformer   | 68.80     | 91.16     | 0.75     | 0.53     | 39.99     | 55.35     | **0.49** | 0.55     |
> > | AirPhyNet   | 70.03     | 94.60     | 0.78     | 0.51     | 43.23     | 58.79     | 0.50     | 0.51     |
> > | Air-DualODE | **66.40** | **90.31** | **0.73** | **0.57** | **39.79** | **54.61** | **0.49** | **0.57** |
> >
> > `Table2` The new sudden change definition based on the **Binary Segmentation detection**:
> >
> > | $\text{PM2.5} \ge 75$μg/m³ | Beijing   |            |          |          | KnowAir   |           |          |          |
> > | -------------------------- | --------- | ---------- | -------- | -------- | --------- | --------- | -------- | -------- |
> > |                            | MAE       | RMSE       | SMAPE    | MDA      | MAE       | RMSE      | SMAPE    | MDA      |
> > | Airformer                  | 85.47     | 105.57     | 0.82     | 0.41     | **50.70** | 66.79     | **0.53** | **0.42** |
> > | AirPhyNet                  | 88.73     | 107.97     | 0.88     | 0.37     | 52.99     | 68.00     | 0.57     | 0.34     |
> > | Air-DualODE                | **84.29** | **105.31** | **0.81** | **0.42** | **50.67** | **66.13** | **0.53** | **0.42** |
> >
> > `Table3` The new sudden change definition based on the **Kernel Change Point Detection**:
> >
> > | $\text{PM2.5} \ge 75$μg/m³ | Beijing   |            |          |          | KnowAir   |           |          |          |
> > | -------------------------- | --------- | ---------- | -------- | -------- | --------- | --------- | -------- | -------- |
> > |                            | MAE       | RMSE       | SMAPE    | MDA      | MAE       | RMSE      | SMAPE    | MDA      |
> > | Airformer                  | 87.96     | **107.54** | 0.84     | 0.44     | 51.74     | 66.95     | **0.50** | **0.41** |
> > | AirPhyNet                  | 91.35     | 110.68     | 0.89     | 0.37     | 52.86     | 67.01     | 0.54     | 0.35     |
> > | Air-DualODE                | **86.80** | **107.51** | **0.83** | **0.45** | **50.98** | **66.79** | **0.50** | **0.41** |
> >
> > From the above three tables, Air-DualODE is indeed better than the baseline models under different definitions of sudden changes and a variety of evaluation metrics.
> >
> >
> >
> > ### **Q2**: Can the authors further discuss the potential threats to validity that may arise when using these metrics and two kinds of evaluation methods?
> >
> > From our perspective, sudden change remains a context-dependent concept and does not have a well-recognized definition in the air quality. Essentially, sudden changes represent a subset of the test set that represent the stakeholder’s interests. The subset defined by stakeholders, based on their specific criteria for sudden changes, often includes challenging or noteworthy scenarios. Based on the results of the three subsets mentioned in Q1, we believe that Air-DualODE demonstrates greater robustness compared to the baselines under five different sudden change settings.

---

> ### Comment · Reviewer_6Tsm · 2024-12-03
>
> I have read your final response. Thank you for addressing my questions and showing dedication until the very end. Your earnest efforts throughout our discussion have been greatly appreciated.​​​​​​​​​​​​​​​​

---

> > ### Author Response · Authors · 2024-12-03
> > **Response to Reviewer 6Tsm**
> >
> > We are happy that you are finally satisfied with our revision, and deeply value your engagement and dedication as a reviewer for ICLR 2025. If our revision fully meets your expectations, would you consider giving us an accept instead of marginally above accept?😉

---

### Meta-Review · Area_Chair_Befo · 2024-12-19

**Metareview:**

This paper presents a spatiotemporal forecasting method specialized to the air quality prediction which is an important problem in some countries. They combine a physics-based modeling and a data-based modeling to have better predictions. However, they failed to consider some recent methods as pointed by the reviewers and during the rebuttal phase, they added more discussion on them. For the physics modeling, they use the diffusion-advection system, which is a quite standard method for this task. Later, they may want to extend to the diffusion-advection-reaction system in my personal opinion. They showed SOTA performance in various datasets.

I recommend accepting this paper, but I also recommend that the authors put more comparisons with more spatiotemporal forecasting baselines. There are many baselines for traffic forecasting. Since they are technically similar to each other, some audiences may be interested in those comparisons.

**Additional Comments On Reviewer Discussion:**

The authors left concrete rebuttal messages and some reviewers are greatly satisfied with them and raised their scores.

---

### Decision · Program_Chairs · 2025-01-22

Accept (Poster)